# BAFopathies' DNA methylation epi-signatures demonstrate diagnostic utility and functional continuum of Coffin–Siris and Nicolaides–Baraitser syndromes

Erfan Aref-Eshghi[1,2], Eric G. Bend[3], Rebecca L. Hood[4], Laila C. Schenkel[1,2], Deanna Alexis Carere[2], Rana Chakrabarti[5], Sandesh C.S. Nagamani[6], Sau Wai Cheung [6], Philippe M. Campeau [7], Chitra Prasad[5], Victoria Mok Siu[5], Lauren Brady[8], Mark A. Tarnopolsky[8], David J. Callen[8], A. Micheil Innes [9], Susan M. White[10], Wendy S. Meschino[11], Andrew Y. Shuen[5], Guillaume Paré[12], Dennis E. Bulman[4], Peter J. Ainsworth[1,2], Hanxin Lin[1,2], David I. Rodenhiser[5,13], Raoul C. Hennekam[14], Kym M. Boycott[4], Charles E. Schwartz[15] & Bekim Sadikovic[1,2]

Coffin–Siris and Nicolaides–Baraitser syndromes (CSS and NCBRS) are Mendelian disorders caused by mutations in subunits of the BAF chromatin remodeling complex. We report overlapping peripheral blood DNA methylation epi-signatures in individuals with various subtypes of CSS (*ARID1B*, *SMARCB1*, and *SMARCA4*) and NCBRS (*SMARCA2*). We demonstrate that the degree of similarity in the epi-signatures of some CSS subtypes and NCBRS can be greater than that within CSS, indicating a link in the functional basis of the two syndromes. We show that chromosome 6q25 microdeletion syndrome, harboring *ARID1B* deletions, exhibits a similar CSS/NCBRS methylation profile. Specificity of this epi-signature was confirmed across a wide range of neurodevelopmental conditions including other chromatin remodeling and epigenetic machinery disorders. We demonstrate that a machine-learning model trained on this DNA methylation profile can resolve ambiguous clinical cases, reclassify those with variants of unknown significance, and identify previously undiagnosed subjects through targeted population screening.

[1] Department of Pathology and Laboratory Medicine, Western University, London N6A 5W9 ON, Canada. [2] Molecular Genetics Laboratory, Molecular Diagnostics Division, London Health Sciences Centre, London N6A 5W9 ON, Canada. [3] Prevention Genetics, Marshfield 54449 WI, USA. [4] Children's Hospital of Eastern Ontario Research Institute, University of Ottawa, Ottawa K1H 8L1 ON, Canada. [5] Children's Health Research Institute, London N6A 5W9 ON, Canada. [6] Department of Molecular and Human Genetics, Baylor College of Medicine, Houston 77030 TX, USA. [7] Department of Pediatrics, University of Montreal, Montreal H3C 3J7 QC, Canada. [8] Department of Pediatrics, McMaster University, Hamilton L8P 1A2 ON, Canada. [9] Department of Medical Genetics, Alberta Children's Hospital Research Institute for Child and Maternal Health, University of Calgary, Calgary T3B 6A8 AB, Canada. [10] Department of Paediatrics, University of Melbourne, Melbourne 3052 VIC, Australia. [11] Genetics Program, North York General Hospital, Toronto M2K 1E1 ON, Canada. [12] Department of Pathology and Molecular Medicine, McMaster University, Hamilton L8P 1A2 ON, Canada. [13] Department of Pediatrics, Biochemistry and Oncology, Western University, London N6A 5W9 ON, Canada. [14] Department of Pediatrics, Academic Medical Center, University of Amsterdam, Amsterdam 1012 WX, The Netherlands. [15] Greenwood Genetic Center, Greenwood 29646 SC, USA. Correspondence and requests for materials should be addressed to B.S. (email: Bekim.Sadikovic@lhsc.on.ca)

BRG1/BRM-associated factor (BAF) is a chromatin remodeling complex which plays an integral role in regulating gene expression, cell differentiation, DNA repair, and neural development[1]. Disruption of the BAF complex has been linked to several neurodevelopmental syndromes, commonly referred to as BAFopathies. Among these are Coffin–Siris and Nicolaides–Baraitser syndromes (CSS and NCBRS), two phenotypically similar genetic conditions characterized by developmental delay and intellectual disability (DD/ID), coarse facial features, and phalangeal abnormalities[2–5]. CSS is genetically heterogeneous and, to date, mutations in more than nine genes have been demonstrated to cause CSS or a CSS-like phenotype. The products of all of these genes are either immediate members of the BAF complex, i.e., ARID1B, ARID1A, SMARCB1, SMARCA4, SMARCE1, ARID2, and DPF2[6,7], or interact with it in the downstream pathways, i.e., SOX11 and PHF6[4,8]. NCBRS is caused by mutations in SMARCA2, encoding another subunit of the BAF complex[3].

Disruption of the BAF complex in CSS/NCBRS may impact DNA methylation by two mechanisms. The first is through a direct interaction between chromatin remodeling and DNA methylation during epigenetic reprogramming[9]. The BAF complex has been shown to influence the establishment of DNA methylation either directly (by recruiting DNA methylases) or indirectly (by altering expression of proteins that regulate DNA methylation)[10,11]. Alternatively, downstream pathways aberrantly impacted by disruption of the complex may result in specific DNA methylation patterns during development[12]. Therefore, we hypothesized that CSS and NCBRS might be associated with specific germ-line DNA methylation profiles in the affected individuals.

We have previously described peripheral blood DNA methylation signatures in multiple neurodevelopmental syndromes resulting from defects in various layers of the epigenetic machinery[13–17], including those involving abnormalities in chromatin remodeling, e.g., Kabuki syndrome[18]. Through these reports, we have also shown that DNA methylation signatures can be used as surrogate markers for molecular diagnostics and disease screening, with performances superior to sequence variant analysis[14–18]. Development of similar approaches may be useful for diagnosing BAFopathies including CSS and NCBRS, in which phenotypic/genotypic heterogeneity hinders clinical/molecular diagnosis.

In the present study, we examine a cohort of CSS/NCBRS patients with mutations in ARID1B (CSS type 1 (CSS1)), SMARCB1 (CSS type 3 (CSS3)), SMARCA4 (CSS type 4 (CSS4)), and SMARCA2 (NCBRS). We describe a DNA methylation signature shared across all of these groups, providing functional evidence linking different subtypes of CSS with NCBRS. We show that this profile is specific to CSS/NCBRS and does not occur across other syndromic DD/ID conditions. As well, we demonstrate that chromosome 6q25 microdeletion syndrome, harboring AR1D1B deletion, encompasses a profile falling within the CSS/NCBRS spectrum. Using this profile, we build a classification algorithm for CSS/NCBRS and demonstrate its ability to resolve ambiguous clinical cases as well as to identify previously undiagnosed subjects through targeted screening of a large cohort of undiagnosed patients with DD/ID.

## Results

### CSS/NCBRS exhibit a DNA methylation epi-signature spectrum. We generated genome-wide methylation data using Infinium methylation arrays from peripheral blood DNA of subjects with both confirmed clinical and molecular diagnosis of CSS/NCBRS. Following quality controls, 399,329 CpG sites (probes)

were retained for analysis. Comparisons were performed separately between patients with CSS1 ($n = 14$), CSS3 ($n = 5$), CSS4 ($n = 2$), and NCBRS ($n = 7$) with age-matched cohorts of controls with a sample size six times larger than each group (Table 1). We identified three probe-sets of 146, 135, and 356 CpG sites with a minimum 10% methylation difference, and a multiple-testing corrected $p$ value < 0.05 (limma multivariable regression modeling), adjusted for blood cell type compositions, for CSS1, CSS3, and NCBRS, respectively (Supplementary Data 1–3). No CpG site met these statistical cut-offs in individuals with CSS4 due to insufficient sample size ($n = 2$). A nominal overlap of 10–20% was observed across the three sets. To examine the extent to which these three probe-sets were specific to every subtype, we performed three separate hierarchical clustering analyses. The first clustering, using probes specific to CSS1, completely separated all of the patients, i.e., CSS1, CSS3, CSS4, and NCBRS, from the controls, with little distinctions across different subtypes (Fig. 1a). The second clustering using the CSS3 probes generated a similar pattern, although it assigned an intermediate cluster to the entire non-CSS3 patients who represented slightly less pronounced methylation changes (Fig. 1b). The NCBRS-specific probes, however, grouped NCBRS and CSS3 into one cluster and CSS1 and controls into the other. The two CSS4 subjects were split between the two clusters (Fig. 2). To assure that these observations are not the result of a general DD/ID methylation pattern or batch effect, a random mix of non-CSS/NCBRS subjects having other forms of DD/ID, assayed on the same batch as the patients, were included in each clustering—none grouped with the CSS/NCBRS samples (Supplementary Figure 1).

These observations indicated that the identified probes are not entirely specific to every BAF subunit and a spectrum of methylation change is shared across different subtypes. To investigate the intra-subtype variability and identify subgroups within these profiles, a consensus clustering analysis was performed on all 29 CSS/NCBRS cases, excluding the controls, using the combination of the identified probes. Following 1000 clustering iterations on 80% subsampling of both probes and subjects at cluster counts ($k$) ranging from 2 to 10, $k = 2$ was found to be the most optimal $k$ with the lowest proportion of ambiguous clustering (PAC) measure[19]. This framework generated two distinct clusters, both with consensuses >0.99 (Fig. 3, Supplementary Figures 2–18). The first cluster was composed of CSS1 subjects, whereas the second cluster grouped patients with CSS3 and NCBRS. The two CSS4 subjects were split between the two clusters. Further iterations at $k > 2$ did not refine the separation of the four subtypes. Only at $k = 7$, the CSS3 patients separated from the NCBRS cohort, at which point the first cluster (CSS1) was subdivided into four groups (Fig. 3, Supplementary Figures 2–18). This suggested that variations among CSS1 subjects are greater than the differences between CSS3 and NCBRS, indicating that while CSS3 and NCBRS have the most similar epi-signatures, they contain methylation changes not present in CSS1 (Fig. 3, Supplementary Figures 2–18). Repeat analyses individually on each of the three probe-sets led to similar observations. We also ensured that the observed patterns are not generated by a bias from differences in the sample sizes or probe counts across the three subtypes (Supplementary Figures 19–21).

### Differentially methylated regions in CSS/NCBRS. The identified probes in the three sets showed a trend towards co-occurrences within genes (Supplementary Data 1–3). Probes found in such genes in one of the three sets were also present to variable degrees in the others (Supplementary Data 1–3). An example includes three probes in the PALM gene (chr19:728040–728385), identified to be hypermethylated in both

**Table 1 Patients with a confirmed diagnosis of CSS/NCBRS**

| ID | Disease subtype (gene) | Nucleotide change[a] | Variant effect | Protein change | Data subset |
|---|---|---|---|---|---|
| MS0674 | CSS1 (*ARID1B*) | c.3444C>G | Nonsense | p.Tyr1148* | Training |
| MS0676 | CSS1 (*ARID1B*) | c.3236delT | Frame-shift | p.Phe1079Serfs*51 | Training |
| MS0678 | CSS1 (*ARID1B*) | c.3716delC | Frame-shift | p.Pro1239Hisfs*5 | Training |
| MS0680 | CSS1 (*ARID1B*) | c.2457dupT | Frame-shift | p.Pro819fs* | Training |
| MS1169 | CSS1 (*ARID1B*) | c.1259delA | Frame-shift | p.Asn420Ilefs*10 | Training |
| MS1170 | CSS1 (*ARID1B*) | c.5605C>T | Frame-shift | p.Gln1869* | Training |
| MS1176 | CSS1 (*ARID1B*) | c.2692C>CT | Frame-shift | p.Arg898* | Training |
| MS1177 | CSS1 (*ARID1B*) | c.3223C>CT | Frame-shift | p.Arg1075* | Training |
| MS1212 | CSS1 (*ARID1B*) | c.3898C>T | Nonsense | p.Gln1300* | Training |
| MS1216 | CSS1 (*ARID1B*) | c.1483C>T | Nonsense | p.Gln495* | Training |
| MS1201 | CSS1 (*ARID1B*) | c.1259delA | Frame-shift | p.N420Ifs*10 | Testing |
| MS1213 | CSS1 (*ARID1B*) | c.1701del | Frame-shift | p.Ala569Profs*21 | Testing |
| MS1215 | CSS1 (*ARID1B*) | c.3352_3359dup | Frame-shift | p.Met1120Ilefs*53 | Testing |
| MS1217 | CSS1 (*ARID1B*) | c.3478delG | Frame-shift | p.Glu1160Argfs*51 | Testing |
| MS0679 | CSS3 (*SMARCB1*) | c.1091_1093delAGA | In-frame deletion | p.Lys364del | Training |
| MS0683 | CSS3 (*SMARCB1*) | c.1130G>A; | Missense | p.Arg377His | Training |
| MS1162 | CSS3 (*SMARCB1*) | c.1121G>A | Missense | p.Arg374Gln | Training |
| MS0681 | CSS3 (*SMARCB1*) | c.1130G>A; | Missense | p.Arg377His | Testing |
| MS1163 | CSS3 (*SMARCB1*) | c.1096C>T | Missense | p.Arg366Cys | Testing |
| MS1168 | CSS4 (*SMARCA4*) | c.1452_1453delGGinsA | Frame-shift | p.Asp485Ilefs*16 | Training |
| MS1209 | CSS4 (*SMARCA4*) | c.2932C>G | Missense | p.Arg978Gly | Training |
| MS1160 | NCBRS (*SMARCA2*) | c.2639C>T | Missense | p.Thr880Ile | Training |
| MS1221 | NCBRS (*SMARCA2*) | c.3602C>T | Missense | p.Ala1201Val | Training |
| MS1223 | NCBRS (*SMARCA2*) | c.3476G>T | Missense | p.Arg1159Leu | Training |
| MS1238 | NCBRS (*SMARCA2*) | c.2642G>T | Missense | p.Gly881Val | Training |
| MS1243 | NCBRS (*SMARCA2*) | c.3404T>C | Missense | p.Leu1135Pro | Training |
| MS1222 | NCBRS (*SMARCA2*) | c.3485G>A | Missense | p.Arg1162His | Testing |
| MS1237 | NCBRS (*SMARCA2*) | c.3475C>G | Missense | p.Arg1159Gly | Testing |

[a]All mutations are in heterozygous state; mean age of CSS1 subjects ($n = 14$): 12.0 ± 7.5, matched controls ($n = 84$): 13.0 ± 9.7; mean age of CSS3 subjects ($n = 5$): 3.0 ± 3.9, matched controls ($n = 30$): 3.0 ± 3.3; mean age of NCBRS subjects ($n = 7$): 11.5 ± 11.3, matched controls ($n = 42$): 11.9 ± 11.2; mean age of patients in the training dataset ($n = 21$): 9.3 ± 8.5, matched controls ($n = 126$): 10.0 ± 8.6

CSS1 and CSS3, two of which were also detected to be hypermethylated in NCBRS (Supplementary Data 1–3). This led us to systematically map differentially methylated regions (DMRs) across the three subtypes and examine the inter-class variations in methylation levels. Using the DMRcate algorithm[20] we found a total of 3, 30, and 24 DMRs with a minimum of three consecutive probes no more than 1 kb apart, >10% average methylation change, and a false discovery rate (FDR) < 0.05 in CSS1, CSS3, and NCBRS, respectively (Supplementary Data 4). Among the three DMRs found in CSS1, a hypermethylated region in chr12:53,343,514–53,343,849, overlapping the *KRT8* promoter and *KRT18* gene body (Fig. 4), met all of the specified cut-offs in all three groups. The other two DMRs in CSS1, although not always reaching the strict cut-offs applied, showed very similar methylation levels in other subtypes (Fig. 5). Almost all of the regions found in CSS3 showed an intermediate pattern in the others, with methylation values between the controls and CSS3 (Supplementary Figure 22). The DMRs found in NCBRS represented the greatest deviation from methylation levels in CSS1 patients, but a striking similarity to those in CSS3 (Figs. 4, 6).

**Shared functionality in the CSS/NCBRS profiles**. To assess what is functionally represented in the methylation profiles, three separate gene-set enrichment analyses were performed for each probe-set. We found 1, 34, and 13 enriched Gene Ontology (GO) terms (FDR < 0.05) for the probe-sets specific to CSS1, CSS3, and NCBRS, respectively (Supplementary Data 5). The single GO term found in CSS1 was anatomical structure morphogenesis. The GO terms in CSS3 were composed of biological cell regulations, locomotion, synapse maturation, embryonic organ development, and regulation of the mitogen-activated protein kinase cascade. The GO terms in NCBRS were mainly related to

anatomical structure development and morphogenesis (Supplementary Data 5, Fig. 7). This indicated that gene networks involved in system development and morphogenesis are represented across all three probe-sets. To improve the power in GO term identifications, the three probe-sets were combined, and the analysis was repeated. We found 191 GO terms composed of anatomical and system development, as the most significant terms, followed by various forms of signaling and cellular process regulations (Supplementary Data 6 and Supplementary Figure 23).

**A classification model for CSS/NCBRS**. These analyses suggest that CSS/NCBRS is associated with a methylation profile which could potentially be used for screening and diagnostic purposes. Due to a significant level of overlap across various CSS/NCBRS groups, we hypothesized that modeling the differences between all CSS/NCBRS patients and controls would provide the greatest power at both stages of feature selection and training. Therefore, data from all 29 patients were combined and randomly divided into two cohorts of training (75% subset, $n = 21$) and testing (25% subset, $n = 8$), ensuring that different subtypes are equally represented in both data subsets (Table 1). A sample of 126 age-matched controls (six times larger than the patient cohort) was added to the training subset for the purposes of feature selection and training. Using moderated $p$ values obtained from the multivariate regression modeling, we prioritized the top 1000 probes that were not confounded by batch effect or blood cell type compositions and showing a minimum 5% detectable methylation change. Feature selection was performed from this list by first measuring the area under the receiver operating characteristic curve (AUC) for every probe to select the most differentiating CpGs (AUC > 0.85), and next by removing the redundant features

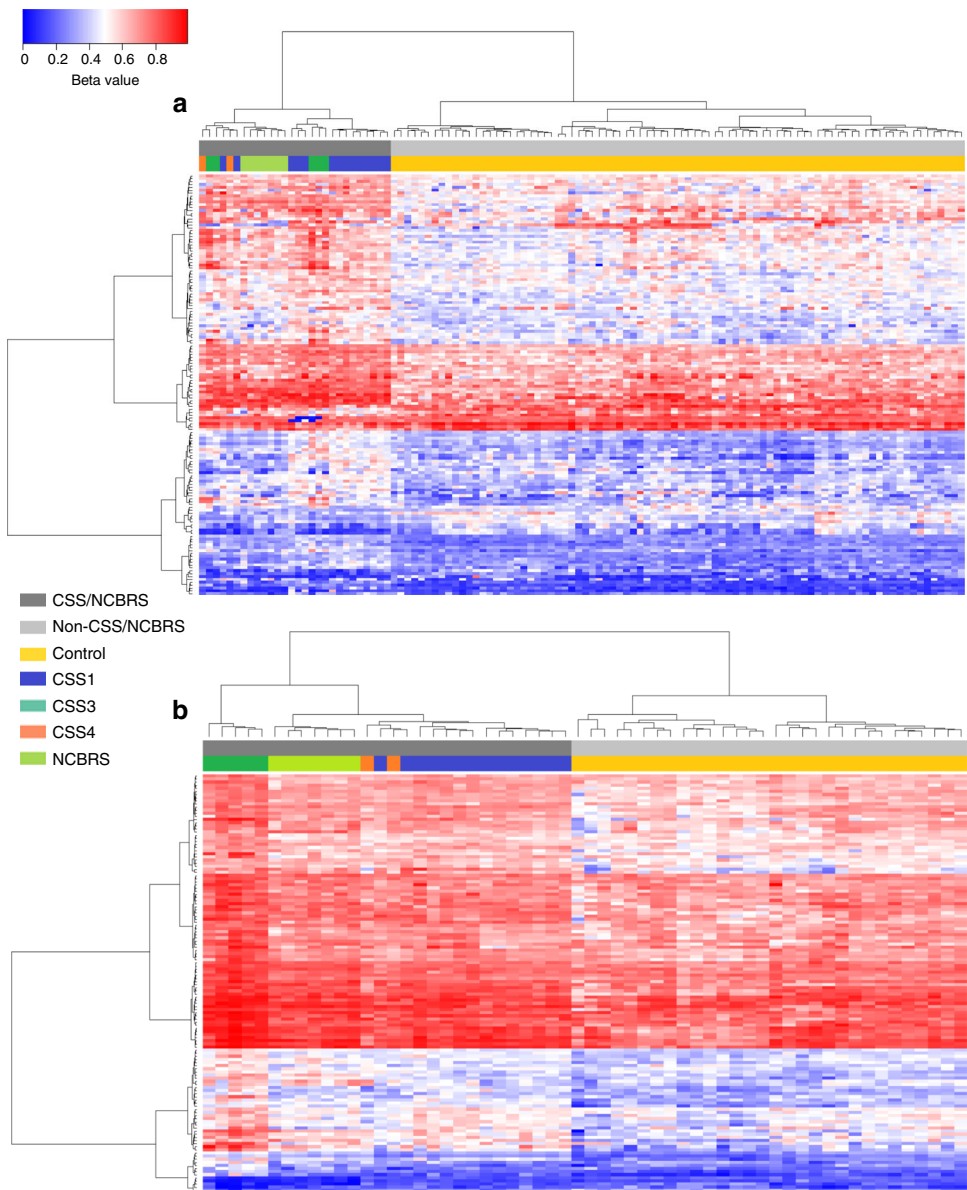

**Fig. 1** Clustering analysis using the probe-sets identified in CSS1 and CSS3. Clustering analysis was performed using Ward's method on Euclidian distance. Rows represent the CpG probes, and columns represent the subjects. The color scale from dark blue to dark red represents the range of the methylation levels (beta values) between 0 and 1. The top pane represents the overall status of the condition, i.e., CSS/NCBRS vs. non-CSS/NCBRS. The lower pane indicates the subtype for every sample including CSS1 (dark blue), CSS3 (dark green), CSS4 (orange), NCBRS (light green), and controls (yellow). **a** Hierarchical clustering of 146 probes differentially methylated between patients with CSS1 and controls generates two clusters: one composed of all CSS and NCBRS patients (CSS1, CSS3, CSS4, and NCBRS), and the other composed of controls and other non-CSS/NCBRS samples. **b** A similar pattern is observed using the 135 probes that were specifically identified for the CSS3 samples; however, patients other than CSS3 (CSS1, CSS4, and NCBRS) show a less pronounced pattern of methylation change, although still being a member of the same CSS3 cluster

with a pairwise correlation coefficient >0.85. The final probe-set ($n = 131$, Supplementary Data 7) was used to train a support vector machine (SVM) with radial basis function kernel on the training cohort. The model was set to generate a score ranging 0–1, representing the probability of a given sample having a methylation profile similar to CSS/NCBRS. A 10-fold internal cross-validation was performed during the training process, which resulted in an accuracy of 98.8% (model details in Supplementary Data 7).

To control for the success of the training procedure, the entire training cohort was supplied to the final model, which assigned correct classifications to all of the cases (scored >0.90) and controls (scored <0.10) used for training (Fig. 8). Next, we confirmed that the model could completely differentiate the non-

CSS/NCBRS samples assayed on the same technical batch of the cases ($n = 244$) from the true CSS/NCBRS samples by supplying their methylation values into the model for prediction. None of these samples were scored high for CSS/NCBRS, indicating the trained model is not sensitive to the batch structure of the data. Additionally, we evaluated the extent to which this model is sensitive to variations in blood cell type compositions by applying it to a total of 60 methylation array data files from six healthy individuals, downloaded from gene expression omnibus (GEO, GSE35069)[21], each being assayed for whole blood, peripheral blood mononuclear cells, and granulocytes, as well as for seven isolated cell populations (CD4+ T, CD8+ T, CD56+ natural killer cell, CD19+ B, CD14+ monocytes, neutrophils, and eosinophils), separately. All of these samples were classified as

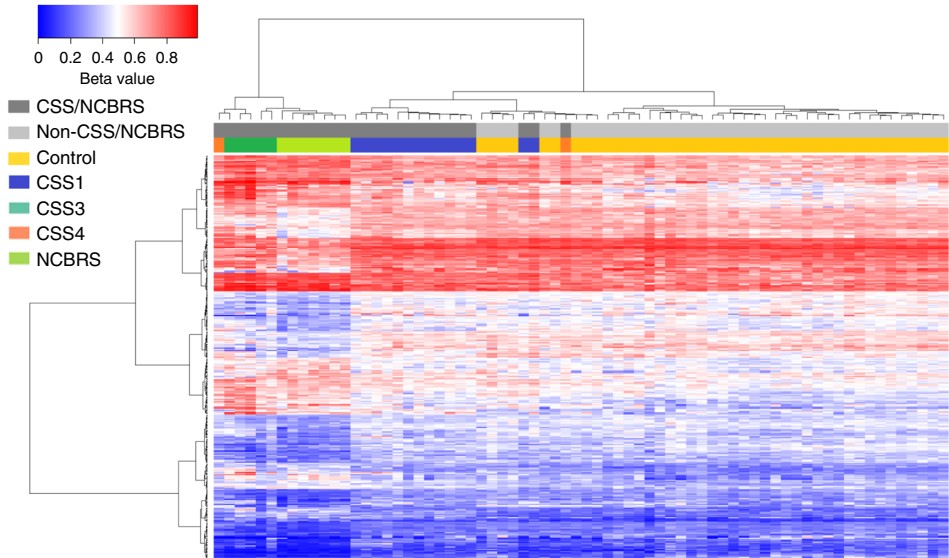

**Fig. 2** Clustering analysis using the probes identified for NCBRS. Clustering analysis was performed using Ward's method on Euclidian distance. Rows represent the CpG probes, and columns represent the subjects. The color scale from dark blue to dark red represents the range of the methylation levels (beta values) between 0 and 1. The top pane represents the overall status of the condition, i.e., CSS/NCBRS vs. non-CSS/NCBRS. The lower pane indicates the subtype for every sample including CSS1 (dark blue), CSS3 (dark green), CSS4 (orange), NCBRS (light green), and controls (yellow). Hierarchical clustering of 365 probes differentially methylated between patients with NCBRS selectively groups CSS3 and NCBRS samples into one cluster, but CSS1 and controls into the other. The two CSS4 subjects are split between the two clusters

non-CSS/NCBRS with scores ranging 0.01–0.13, consistent with those generated for the whole blood samples. The average inter-cell-type variability in the scores was determined to be not more than 10% (Supplementary Data 8).

Next, we applied the model to the CSS/NCBRS samples in the testing cohort, composed of 8 subjects (4 CSS1, 2 CSS3, and 2 NCBRS) who were not used for feature selection or model training. All of these samples were scored >0.90 (Fig. 8), confirming that the model can detect the affected subjects. To measure the specificity of our classifier, we tested 122 healthy subjects with various racial backgrounds (aged 0–36 years) from our internal dataset, combined with two publically available whole blood methylation cohorts from healthy subjects, downloaded from GEO, including 48 children aged 6–14 years (GSE104812)[22], and 186 adults aged 25–67 years (GSE67705)[23]. All of these 356 healthy subjects received very low scores (<0.2) for CSS/NCBRS (Fig. 8). We also questioned whether the model could differentiate the CSS/NCBRS patients from other forms of DD/ID or other Mendelian conditions that result from defects in the epigenomic machinery. DNA methylation profiles of a total of 531 subjects diagnosed with 20 different such syndromic conditions (details in Methods) were supplied to the model for classification, all of which were scored very low for CSS/NCBRS, suggesting that the profile of CSS/NCBRS does not overlap with other DD/ID syndromes.

Due to the relatively small sample sizes and incomplete distinctions across various CSS/NCBRS subtypes, as well as lack of data on other genes involved, a supervised stratification of BAFopathies into their subtypes was not possible. However, for clinical purposes, it is desirable to pinpoint the causative genes. To this end, any subject that was predicted using our classification model to be a case of CSS/NCBRS was also assessed using the consensus clustering analysis (Fig. 3) to determine which of the two previously identified clusters (CSS1 vs. CSS3-NCBRS) the sample segregates with, enabling us to further narrow down the genetic etiology.

**Chr6q25 microdeletion syndrome and CSS/NCBRS profile.** We first used the classification model to assess four subjects with

chromosome 6q25 microdeletion syndrome (Table 2). This condition results from variable length deletions in the interstitial region of the long arm of chromosome 6 in different individuals. The shortest of these was recently mapped in one patient to a 1.1 Mb region containing only the *ARID1B* gene[24]. Despite the suggestion that *ARID1B* is the key gene in this syndrome, clinical presentations are not entirely similar to CSS and tend to diverge as the deletion coordinates varies. Our model assigned scores >0.95 (Fig. 8) to all four patients with Chr6q25 deletion syndrome (deletions ranging 3.7–13.8 Mb in size, all overlapping *ARID1B*), suggesting that this syndrome falls within the spectrum of the CSS/NCBRS methylation profile. With the initial assumption that haploinsufficiency of *ARID1B* generates a pattern similar to CSS1, we assessed these patients using the consensus clustering. In contrast to our expectation, we observed that only the sample with the shortest deleted segment (3.7 Mb) was assigned to the CSS1 cluster, while the rest were clustered with the CSS3-NCBRS group (Table 2). This indicates that as the size of the deletion is increased, the subjects may obtain additional methylation changes not present in CSS1. It may also suggest that the methylation profile spectrum of BAFopathies is not completely stratified by the causative genes.

**Classification of cases with uncertain CSS/NCBRS diagnosis.** To evaluate the utility of this model in clinical molecular diagnostics, we attempted to classify a total of 18 subjects for whom the diagnosis was uncertain/unconfirmed or who carried variants of unknown clinical significance (VUS) in a CSS/NCBRS-related gene (Table 2). We first tested two subjects with clinical suspicion for CSS, one carrying a de novo VUS in *ARID1B* (c.5833T>C, p. Cys1945Arg) and the other one with no sequence data available at the time of assessment. Both of these patients were predicted to have CSS/NCBRS by our model (scored 0.88 and 0.94) and were assigned CSS1 category by the consensus clustering. In silico analysis of the missense variant in the first patient revealed that it is conserved at both nucleotide and amino acid levels, located in an evolutionarily conserved gene segment, and is not found in any public genetic variant database, complying with our

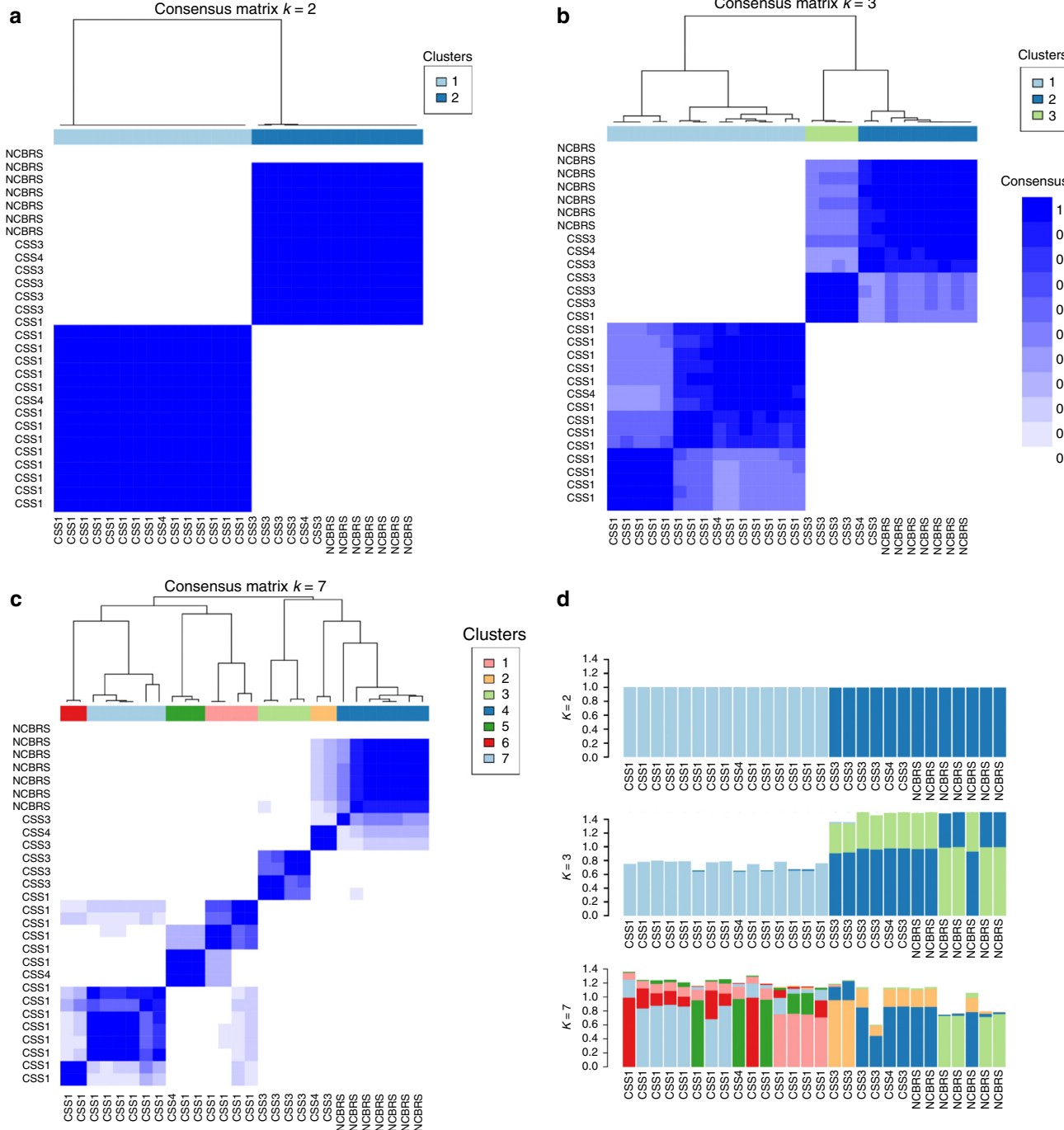

**Fig. 3** Consensus clustering analysis of CSS/NCBRS patients. A 1000 repeat iterations of *K*-means clustering on Spearman's distance was performed, each for a random 80% subsampling of all significant probes and patients, for clustering counts (*k*) 2–10. The most stable cluster (optimal *k*) was determined using the PAC measure as two (Supplementary Figure 12). **a**, **b**, **c** Consensus cluster heatmaps at *k* = 2, 3, and 7. Both rows and columns are samples. The intensity of blue represents how often every two samples from the rows and columns cluster together in the 1000 repeat iterations. The pane on top demonstrates the consensus cluster for each sample. At *k* = 2, CSS3 and NCBRS samples always cluster together while they never group with CSS1 samples. As the *k* increases, more clusters are identified within the initial two groups. At *k* = 7 (**c**), CSS3 subjects for the first time belong to clusters with no NCBRS membership, while at the same time a larger number of clusters are formed within the initial CCS1 class. **d** Item-consensus plots for heatmaps in **a**–**c**. Every subject has an item-consensus value for each of the identified clusters at different *k*s, as indicated by the segmental lengths of the bars, representing the level of consensus of that sample with the members of other clusters (indicated by cluster label colors). The plot at *k* = 2 shows that none of the two clusters contain any consensus with members of the other, being very pure members of their own clusters. As the *k* increases, the purity is reduced. At *k* = 7, all of the samples in NCBRS cluster (dark blue) harbor >20% consensus with the two samples in the yellow cluster (composed of CSS3 and CSS4). These two samples also show the same amount of consensus with NCBRS members, but they have very little consensus with the light green cluster, composed of other CSS3 samples, suggesting that these clusters do not represent differences between CSS3 and NCBRS. None of these patterns are related to batch effect. The three *k*s represented here were chosen as they harbor subdivisions within the CSS-NCBRS cluster. For complete data see Supplementary Figures 2–18

prediction. Targeted exome sequencing of ~4600 Mendelian genes, including *ARID1A*, *ARID1B*, *ARID2*, *SMARCB1*, *SMARCA4*, and *SMARCA2* in the second subject, identified an in-frame duplication in *ARID1A* (c.6507_6509dup), not reported in any public variant database. While this variant remains to be of unknown significance, the involvement of other CSS-related genes cannot be completely ruled out.

Next, we evaluated one subject with a VUS in *SMARCA2* (c.2647C>G, p.Pro883Ala) at a codon in which alternative amino acid changes were known to cause NCBRS. This patient was considered to be a good clinical match for NCBRS, scored 0.83 by our model, and was assigned to the CSS3-NCBRS class by the consensus clustering. Parental testing later confirmed a de novo inheritance, leading to a change in the variant classification from VUS to likely pathogenic. Subsequent trials were performed on two subjects with likely pathogenic variants in *SMARCA2*, as determined according to the American College of Medical Genetics (ACMG) guidelines for classification of sequence variants[25], but with no clinical evaluation reports available at the time of the study. One of these subjects was scored 0.71 and assigned CSS-NCBRS category. She carried a novel de novo missense change (c.2261G>C) in a highly conserved nucleotide which was predicted by multiple in silico assessments to be deleterious. Subsequent clinical assessments were found in favor of an NCBRS diagnosis. The second case had a novel de novo missense change (c.1697C>T). Strict adherence to the ACMG guidelines resulted initially in a likely pathogenic variant classification (PS2, PM2, and PP2). However, clinical evaluations demonstrated that the patient does not exhibit features consistent with NCBRS, and in silico analysis did not conclude a deleterious effect for the variant. Of interest, this subject was scored <0.01 by our classifier. This latter finding was considered as strong functional evidence for benign variant effect (BS3), which resulted in downgrading this variant to likely benign.

Finally, we screened 11 subjects with VUSs identified by previous *ARID1B* testing, without any prioritization based on clinical data. None of these were predicted to have a profile similar to CSS/NCBRS by our algorithm (Table 2, all scored <0.05). This finding is consistent with the fact that non-truncating variants in *ARID1B* are rarely pathogenic in CSS. Also, we tested two subjects, with missense VUSs in *ARID1A*, both of which were scored low by our model (0.06 and 0.17). We did not have a CSS case resulting from *ARID1A* mutations in our training dataset, and thus cannot assume a confident prediction on subjects carrying *ARID1A* variants. However, most pathogenic *ARID1A* variants in CSS are truncating, the two patients tested did not have a typical presentation of CSS, and both of the reported variants are present in public databases with allele frequencies >0.0001. Therefore, our model predictions for these two subjects seem to be consistent with the available information.

**Targeted screening of undiagnosed DD/ID patients**. Next, we addressed the utility of this model as a screening method for identification of new CSS/NCBRS cases in a targeted population. We applied the algorithm to a total of 508 samples from individuals affected by various forms of DD/IDs, dysmorphic facial features, and other miscellaneous clinical findings, but without a conclusive clinical/molecular diagnosis. From this cohort, our model classified two subjects as CSS/NCBRS with scores 0.98 and 0.86 (Fig. 8). The first subject clustered with the CSS1 group, while the other was assigned a CSS3-NCBRS cluster by the consensus clustering. Evaluations of the medical records identified the first case as a 6-year-old male presenting with global developmental delay, coarse facial features, flattened nose, sparse hair, first finger clinodactyly, mild ptosis, and seizures. Brain

imaging identified a Chiari malformation Type I. Subsequent exome sequencing identified two frame-shift mutations in *ARID1B* in close proximity (c.5151del and c.5153del), both resulting in a premature stop codon in the 48th amino acid downstream of the mutation (p.Lys1718Argfs*48). These mutations together accounted for 50% of the sequencing reads (33% and 16% each), suggesting a heterozygous mosaic status, which was later supported by Sanger sequencing. The mutation screening combined with the clinical phenotype assessment confirmed this patient's diagnosis of CSS1. The second subject was a 2-month-old male with small hands and ears, hypospadias, and asymmetry of palpebral fissures. Chromosomal microarray testing was normal and sequence variant analysis of ~4600 clinically relevant genes, including *ARID1A*, *ARID1B*, *ARID2*, *SMARCB1*, *SMARCA4*, and *SMARCA2*, did not identify a possible causative variant. Analysis of indels and copy number variations, obtained from the exome data, also did not find a pathogenic event in a gene known to cause a DD/ID condition. These observations indicated that our method could be applied to screening and identification of undiagnosed cases of CSS/NCBRS or to suggest a diagnosis when a causative sequence variant cannot be identified by standard gene sequencing.

## Discussion

Since the report of the Coffin–Siris syndrome in 1970[2], evidence has accumulated to consider BAFopathies as a continuum. This was first established by the clinical similarity, next by a shared genetic etiology, and here by identification of overlapping DNA methylation signatures. The two BAFopathies presented here, CSS and NCBRS, constitute a phenotypic continuum with roughly defined separation. Both syndromes represent DD/ID, sparse hair, dysmorphic facial features, seizures, and phalangeal abnormalities. The main distinction between them is made by the distal limb features[5–8]. Typical NCBRS patients represent prominent interphalangeal joints and distal phalanges, whereas CSS cases mainly show hypoplasia/aplasia of the fifth fingers/fingernails[26]. However, such distal limb abnormalities are variably presented among the patients, ranging from a complete absence of the fifth finger in some CSS patients with *ARID1A* mutations to only fingernail hypoplasia in the others[27]. Notably, in up to 25% of the CSS cases, no distal limb abnormality is observed[27]. Similarly, while not all NCBRS cases present with prominent interphalangeal joints, this hallmark of NCBRS is found in some CSS3 and CSS4 subjects[28]. As a result, CSS and NCBRS cases have been frequently diagnosed as one another[4,29]. Recently, two cases with duplications of *SMARCA2* were described whose phenotypic features more closely resembled CSS rather than NCBRS[30]. Wieczorek et al.[4], following a comprehensive molecular and clinical assessment of CSS and NCBRS, reported that some of the patients represented an intermediate phenotype between CSS and NCBRS, proposing that these syndromes may represent a disease spectrum rather than two distinct disorders.

Our findings are consistent with these reports. We showed that the DNA methylation profile identified in CSS1 differentiates not only CSS1 subjects, but also CSS3, CSS4, and NCBRS cases from controls, suggesting a shared DNA methylation landscape across all CSS/NCBRS subtypes. Separate analyses for CSS3 and NCBRS showed that these two subtypes have additional methylation changes absent from CSS1, yet they are not easily distinguishable from each other. This indicates that a stronger similarity can exist between some CSS subtypes with NCBRS than there is within CSS. However, DNA methylation alone may not elucidate the entire functional basis of CSS/NCBRS to allow for unification of the two syndromes. Also, using our retrospective study design, it is difficult to decipher whether these changes are causal/

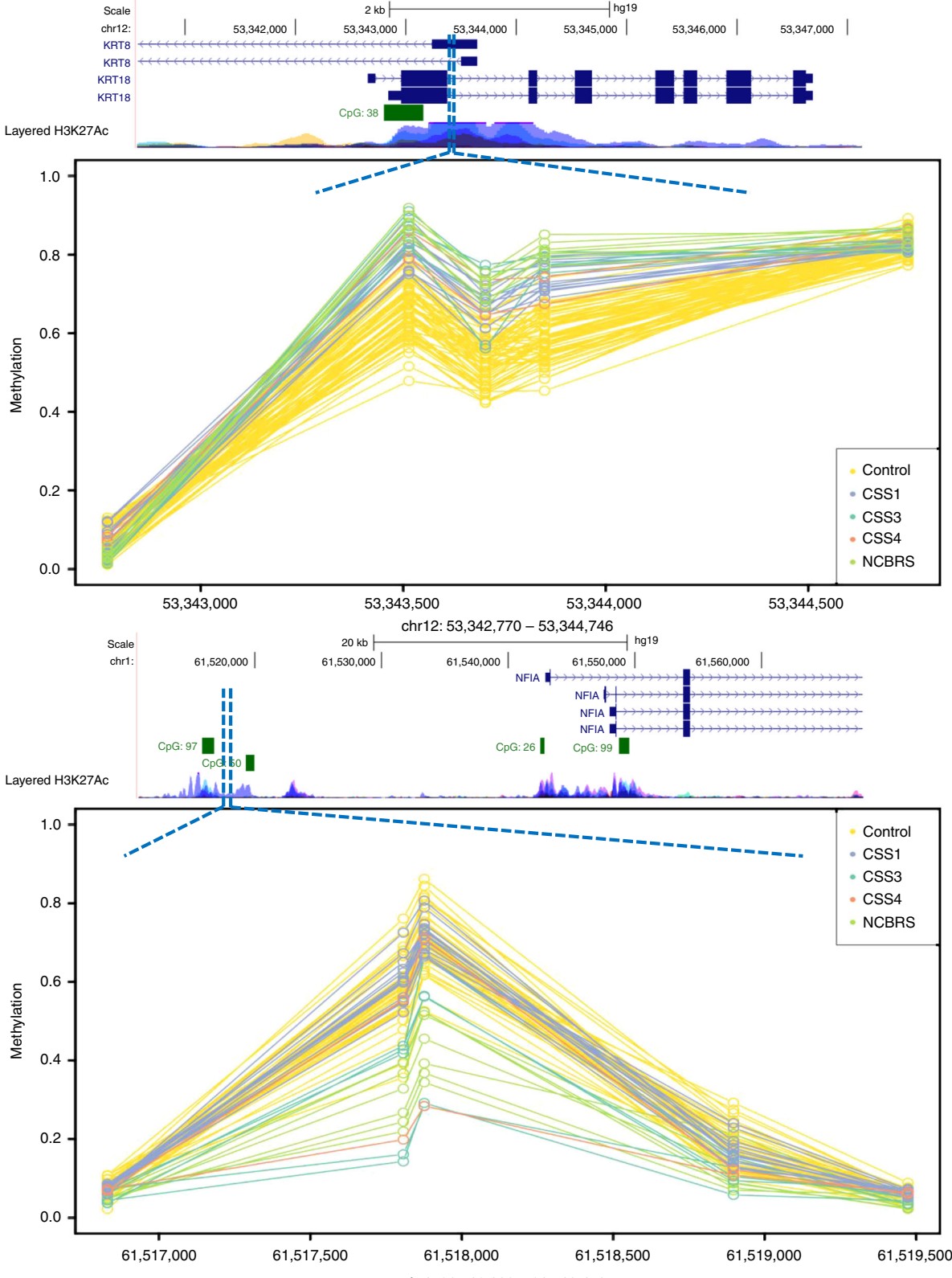

**Fig. 4** Two genomic regions differentially methylated in CSS/NCBRS The top panel illustrates three consecutive probes annotating to the promoter of the *KRT8* gene and the first and second intron of two isoforms of *KRT18*. The region is located in a segment highly enriched for H3K27 acetylation, a marker of active promoters. The methylation levels for every CpG (0–1) is shown using a circle, connected to the adjacent CpGs of the same individual using a line. Control subjects (yellow) show a hemi-methylation pattern in this region. All of the CSS/NCBRS cases show an increase in the methylation level of this segment. The lower panel depicts a segment upstream *NFIA*, located in a CpG island shore region with active H3K27 acetylation, potentially an active promoter. The region is hypomethylated in the CSS3 and NCBRS cases as compared to the controls; however, no difference is observed between CSS1 and controls for this segment. It is noticeable that one of the two CSS4 subjects shows a hypomethylation pattern similar to CSS3 and NCBRS, whereas the other one tends to be similar to controls and CSS1 subjects

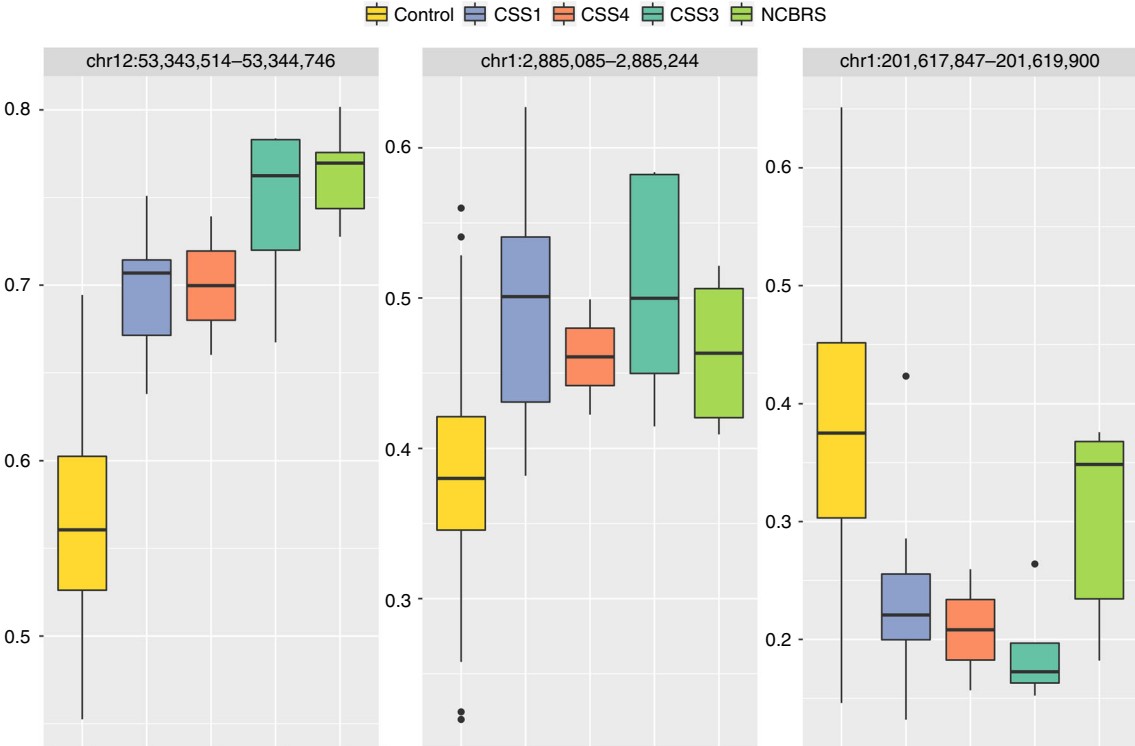

**Fig. 5** Genomic regions differentially methylated in CSS1. The figure shows the entire three DMRs found to be differentially methylated in CSS1 cases compared to the controls. The box plots represent the distribution of median methylation values across all of the probes mapping to each region as stratified by the subtypes, i.e., Controls, CSS1, CSS4, CSS3, and NCBRS. These regions are detected in the comparison between CSS1 and controls; however, all of them show a similar pattern of methylation in all other CSS/NCBRS subtypes. Center line: median of regional methylation levels across samples; lower and upper bounds: first and third quartiles; whiskers: interquartile ranges

consequential, or determine at what stage they occur during the cascade of events happening between the dysfunction of the BAF complex and establishment of the neurodevelopmental phenotypes. Since the observed changes are small in effect size and are restricted to <0.1% of the entire CpGs tested across the genome, it is unlikely that they represent any fundamental event at the single-gene regulatory level or during epigenomic reprogramming.

Notwithstanding, the identified methylation changes occur in genes that are highly enriched in developmental, morphogenesis, and regulatory pathways. In particular, the profile contains multiple CpGs located in the genes involved in cell signaling and neural differentiation. One example is the *PALM* gene which encodes a member of the Paralemmin protein family involved in plasma membrane dynamics in neurons[31]. A second example is the *KRT8* promoter (Fig. 4), the most consistently differentially methylated region in this study (hypermethylated in all CSS/NCBRS subjects). KRT8 is a type II keratin family member which acts as an intermediate filament in epithelial cells and plays a role in maintaining the integrity of the cell structure, signal transduction, and cellular differentiation[32]. Also, the overall pattern of the epi-signatures seems to be consistent with the phenotypic variations across CSS/NCBRS. CSS3 patients represent the most severe form of the disease, whereas CSS1 cases typically exhibit the milder end of the spectrum and often have normal growth[4,27,33]. Consistent with this, we observed that the majority of shared methylation components across CSS/NCBRS are most exaggerated in CSS3, while methylation changes observed in CSS1 seem to harbor the most moderate effect sizes of all. Of interest, the DNA methylation similarity between CSS3, NCBRS, and one of the CSS4 subjects may correlate with the occasional occurrence of interphalangeal joint prominence in NCBRS, CSS3,

and CSS4, but not in CSS1. These phenotypic correlations together with the over-representation of the developmental biological processes in the methylation signatures can reflect the more downstream biological pathways that might be influenced by a disruption of the BAF complex[1].

While the degree of overlap in the epi-signatures extends beyond CSS/NCBBRS to Chr6q25 microdeletion syndrome, the observed changes remain to be limited to BAFopathies and do not occur in a similar pattern in other neurodevelopmental disorders or those conditions caused by defects in other epigenetic regulators. Different histone modifying enzymes are known to affect shared pathways which might result in overlapping epigenetic changes across the chromatin-related disorders. We addressed this issue in CSS/NCBRS by supplying methylation values from a large cohort of such conditions ($n = 531$) to our classification model. All of these subjects received very low scores for CSS/NCBRS, suggesting that despite overlapping targets in the histone modifiers, the downstream changes are unique to each condition, indicating that the DNA methylation signature of CSS/NCBRS could have an invaluable clinical utility.

We took advantage of these findings and developed a classification tool to be used for clinical diagnosis and screening. Using multiple external and internal datasets we confirmed that our model is completely specific to CSS/NCBRS, and its performance is not influenced by variations in sex, age, ethnicity, and blood cell type compositions, or does not falsely detect other developmental conditions. The rarity of CSS and NCBRS (~200 CSS and ~100 NCBRS cases reported worldwide—considered to be an underestimate)[28,34], overlapping clinical features with other conditions, and non-specific presentations limit the clinical detection of these syndromes. The CSS1 patient we identified through targeted screening had remained undiagnosed for years, despite having a

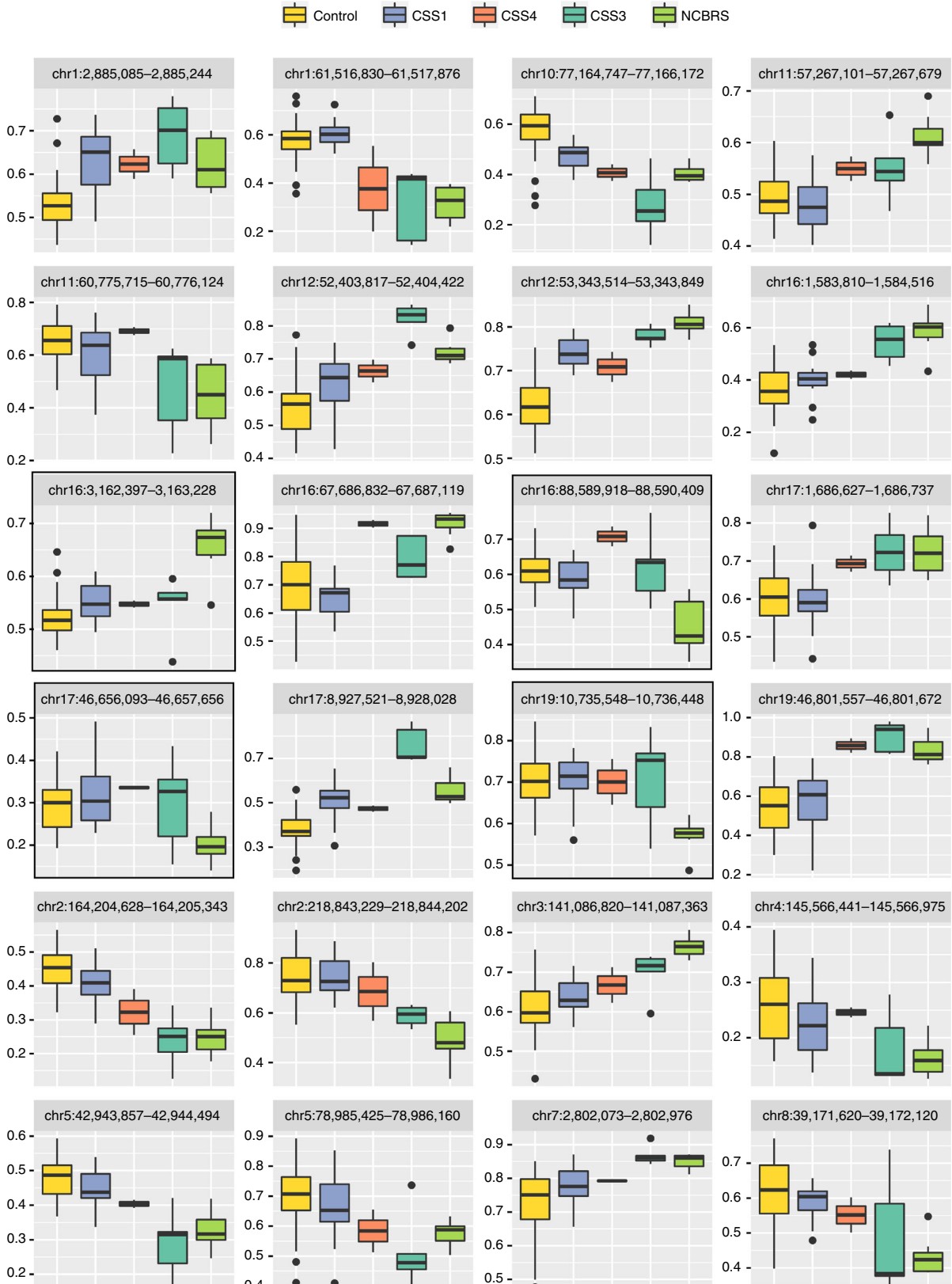

**Fig. 6** DMRs differentially methylated in NCBRS. The figure shows the entire 24 DMRs found to be differentially methylated in NCBRS cases compared to controls. The box plots represent the distribution of median methylation values across all of the probes mapping to each region as stratified by the subtypes, i.e., Controls, CSS1, CSS4, CSS3, and NCBRS. These regions are detected in the comparison between NCBRS and controls; however, except for four regions (outlined by black squares), all show a similar pattern of hypo/hypermethylation in the CSS3 category. In most cases, when this happens, the identified trend is more pronounced in CSS3. The CSS1 samples, however, tend to represent the end of a methylation spectrum that is more similar to controls. Center line: median of regional methylation levels across samples; lower and upper bounds: first and third quartiles; whiskers: interquartile ranges

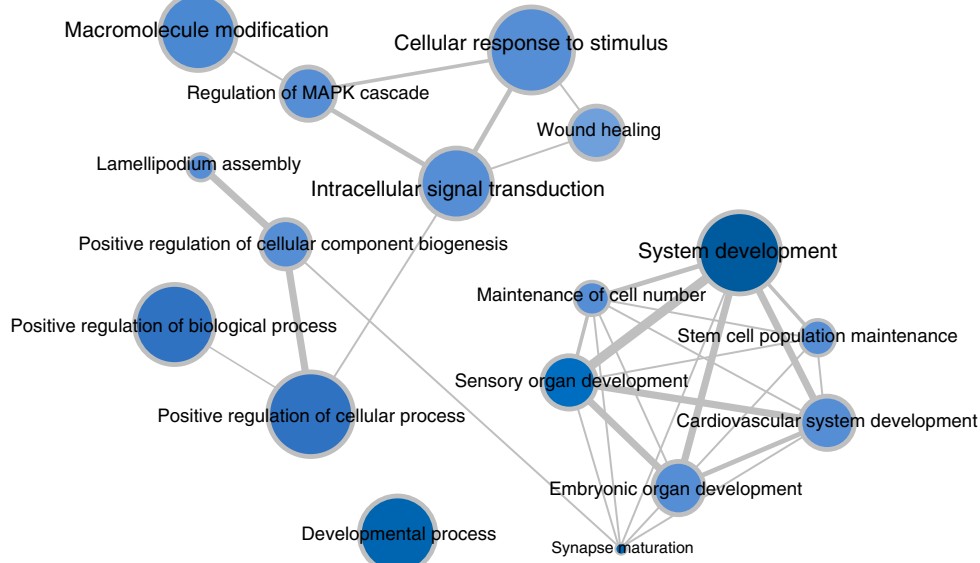

**Fig. 7** Gene ontology terms enriched in the CSS/NCBRS methylation profile. The figure is generated from a reduced list (redundant GO terms have been removed) of the entire 58 GO terms found to be significantly enriched, separately, in the three probe-sets related to CSS1, CSS3, and NCBRS (Supplementary Data 5). The color intensity represents the degree of significance (*p* value obtained from a hypergeometric test conducted by missMethyl package). The size of each circle indicates the number of GO terms it was related to before the reduction of the redundant terms was performed. The thickness of the connecting lines corresponds to the level of interactions and relatedness among the pathways. System development followed by regulation of cellular process represent the most significant GO terms in this analysis. Visualization of the 192 GO terms, found after combining the three probe-sets (Supplementary Data 6), is illustrated in Supplementary Figure 23

relatively typical presentation of the syndrome. Notably, he did not have hypoplasia of the fifth finger, a clinical hallmark for CSS diagnosis. Such a diagnosis is further challenged by the fact that DNA sequence analysis cannot always provide a conclusive result. For roughly 40% of the CSS cases reported in the literature, a molecular diagnosis has not been established[28,34]. While some of these patients might carry mutations in yet-to-be identified genes, others carry VUSs, or variants not sequenced in routine screening, such as large indels or noncoding variants. We demonstrate that DNA methylation profiling provides a powerful solution to these challenges. When a non-CSS/NCBRS classification is made for a subject with a VUS, this tool can confidently rule out a deleterious effect for the suspected variant. An example was the subject with a de novo missense variant in *SMARCA2* (Table 2), where strict adherence to the variant classification system had resulted in a likely pathogenic classification. This patient was found later to not show either the clinical features or the methylation profile consistent with NCBRS, indicating that the reported variant was benign and sequence variant assessment alone without consideration of clinical and functional data can sometimes be misleading. It is worthwhile, however, to note that a CSS/NCBRS classification may not necessarily indicate a pathogenic effect for a variant, since unreported sequence variants may be responsible for the methylation signature. Additionally, we showed that our method could diagnose BAFopathy cases, for whom a causative variant cannot be found, a very common phenomenon in CSS[28,34]. We propose that genomic DNA methylation assessment has a potential to become part of the clinical screening of patients with broad ranges of developmental disorders, taking into consideration that, in addition to BAFopathies, it allows for concurrent assessment of multiple syndromes resulting from DNA methylation defects, including imprinting conditions[35], Fragile X syndrome[36], and other previously described conditions with epi-signatures[17]. Given the cost of DNA methylation microarray testing is similar

to the cost of a single-gene test, it has a potential to be adapted in molecular diagnosis together with current genomic screening tests.

Alongside unraveling the functional continuum and diagnostic utility of an epigenetic signature in BAFopathies, our findings generate new questions to be explored by further research. We have not investigated whether the observed changes are present in other organs, particularly in the neural tissue, the main source of involvement in BAFopathies. A previous study has shown that a hypomethylation profile detected in the blood of the Sotos syndrome can also be found in the fibroblasts of the patients; yet, this might not be the case for BAFopathies. In addition, the impact of the observed methylation signatures on the pathology of the disease is unknown. These may involve integrative investigation of gene expression changes and histone modification patterns among the affected individuals. Animal models to study development can reveal the origin of the changes and potentially shed light on the pathways involved in pathogenesis. Other less common gene mutations associated with BAFopathies will need to be investigated. With more samples it may be possible to examine differences in methylation patterns across different mutation types, e.g., loss vs. gain of function, and different genes encoding different proteins of the BAF complex.

DNA methylation testing in unsolved DD/ID cohorts has significant potential, in conjunction with sequence variant analysis, to enable new molecular diagnoses. Gene sequencing is expected to remain the gold standard for the diagnosis of Mendelian conditions; however, in many cases, it will need to be augmented with functional evidence. Similarly, description of Mendelian diseases and delineation of the phenotypes will be influenced by the functional data. This report presents an example of how functional evidence, obtained through high-throughput epigenomic and computational approaches, can change our understanding of human disease and provide us with more efficient tools for screening and diagnosis.

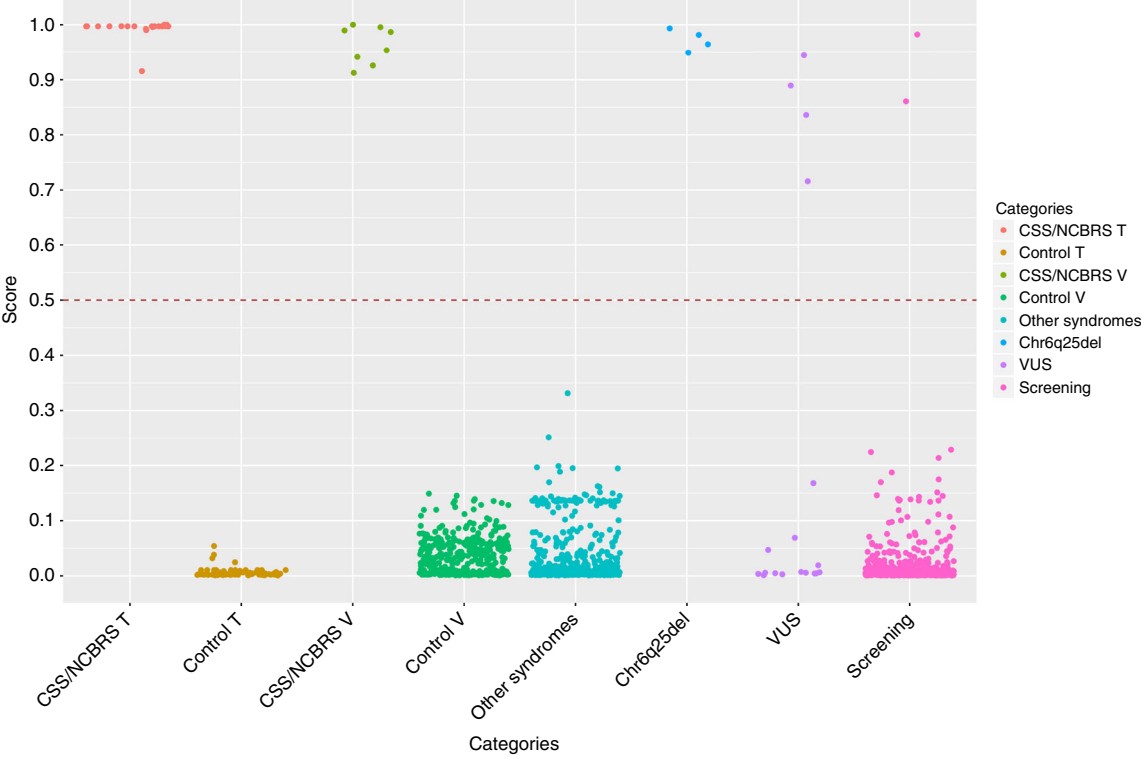

**Fig. 8** Scores generated for different subjects by the CSS/NCBRS classification model. An SVM classifier generates scores for every subject as the probability of having a DNA methylation profile similar to what is observed in CSS/NCBRS. The y axis represents scores 0–1, with higher scores indicating a higher chance of carrying a methylation profile related to CSS/NCBRS, as stratified on the x axis for different groups of tested subjects. Every point represents a single sample. By default, the SVM classifier defines a cut-off of 0.5 for assigning the class (dashed line); however, the vast majority of the tested individuals received a score <0.2 or >0.8. Therefore, to improve visualization, the points are jittered. CSS/NCBRS T: CSS/NCBRS patients from the training set (n = 21); Control T: control samples used to train the model (n = 126); CSS/NCBRS V: Samples from the testing set (n = 8), who were not used for feature selection or model training; Control V: healthy subjects from one internal (n = 122) and two external cohorts (n = 48 and 186) used to measure the specificity of the model (n = 356); Other syndromes: subjects diagnosed with various syndromic diseases presenting with DD/ID (n = 531, details in Methods); Chr6q25del: four patients with interstitial deletions in Chr6q25 (Table 2); VUS: subjects with variants of unknown clinical significance (VUS) in a CSS/NCBRS-related gene, or with clinical suspicion for CSS/NCBRS (Table 2); Screening: subjects with various presentations of developmental delay and intellectual disability, but with no diagnosis, used for case finding (n = 508). The first two categories represent the classification of the subjects used to train the algorithm. The second two validate the performance of the classification model on the testing dataset. In the fifth category, the model demonstrates the ability to accurately distinguish other DD/ID cases from CSS/NCBRS. All of the Chr6q25del patients (sixth category) are scored high for having a profile related to CSS/NCBRS. The model classifies four of the subjects with VUS or clinical suspicion for CSS/NCBRS as a case of CSS/NCBRS and assigns non-CSS/NCBRS status to the rest of the subjects in the seventh category. In the last category, screening of a DD/ID cohort identifies two subjects as a potential case of CSS/NCBRS

## Methods

**Patients and cohorts.** Peripheral blood DNA samples from patients with CSS were collected from the Greenwood Genetic Center (Greenwood, SC, USA), London Health Sciences Center (London, ON, Canada), and Care4Rare Canada Consortium (CHEO Research Institute, Ottawa, ON, Canada). Subjects diagnosed with NCBRS were recruited from the Department of Pediatrics at the University of Amsterdam (Amsterdam, The Netherlands). These cases were previously described in detail by Van Houdt et al.[3]. Patients with Chr6q25 deletion syndrome were recruited from Baylor College of Medicine (Houston, TX, USA), previously delineated clinically/molecularly in detail by Nagamani et al.[37].

Control subjects were selected from our laboratory reference cohort that is composed of individuals without any abnormal imprinting defects or known diseases that influence the epigenomic profile. This reference cohort was previously preselected from a larger cohort of approximately 1000 individuals across a broad range of age, sex, and ethnicity distribution. The validation cohort, which was later used to measure the specificity of our algorithm, is composed of healthy individuals collected from the Greenwood Genetic Center, and two publically available databases obtained from GEO (GSE67705 and GSE104812)[22,23].

Samples from patients with other diagnosed DD/ID conditions were collected from Greenwood Genetic Center and Care4Rare Canada Consortium. Patients with Angelman syndrome, Prader–Willi syndrome, Beckwith–Wiedemann syndrome, Sotos syndrome, Claes–Jensen syndrome, CHARGE syndrome, Kabuki syndrome, ATRX, Coffin–Lowry syndrome, Rett syndrome, Fragile X syndrome, autism spectrum disorders, and RASopathies were recruited from the Greenwood Genetic Center. Peripheral blood DNA samples from patients with autosomal dominant cerebellar ataxia with deafness and narcolepsy, Genitopatellar syndrome,

and Floating–Harbor syndrome were collected from the Care4Rare Canada Consortium. Subjects with trisomy 21 were recruited from McMaster University Medical Centre (Hamilton, ON, Canada). Samples with Silver–Russell syndrome were downloaded from GEO (GSE104451 and GSE55491)[38,39].

Any subject used herein to represent a condition had a confirmed clinical diagnosis of the aforementioned syndrome and was screened for mutations in the related genes. The mutation report from every patient was reviewed according to the ACMG guidelines for interpretation of genomic sequence variants[25], and only individuals confirmed to carry a pathogenic or likely pathogenic mutation together with the clinical diagnosis were used to identify an epi-signature or to represent a syndrome.

Samples with uncertain clinical/molecular diagnosis for CSS/NCBRS were collected from Greenwood Genetic Center, London Health Sciences Center, and Care4Rare Canada Consortium. Targeted screening was performed in a cohort composed of patients with DD/ID but with no clinical/molecular diagnosis at the time of recruitment. These patients have been recruited on an ongoing basis since 2014 from the Department of Pediatrics at McMaster University and London Health Sciences Centre, to be part of an ongoing project aiming at the development of a genome-wide DNA methylation testing for DD/ID conditions.

**Methylation assay and quality control.** Peripheral whole blood DNA was extracted using standard techniques. Following bisulfite conversion, DNA methylation analysis of the samples was performed using the Illumina Infinium methylation 450k or EPIC bead chip arrays (San Diego, CA), according to the manufacturer's protocol. The resulting methylated and unmethylated signal intensity data were imported into R 3.4.2 for analysis. Normalization was

**Table 2 Classification of subjects with Chr6q25 deletions, with variants of unknown significance in a CSS/NCBRS-related gene, or with a clinical suspicion for CSS/NCBRS**

| ID | Suspected genetic change | In silico assessment[a] | Variant population allele frequency | Primary ACMG variant classification | CSS/NCBRS score (Consensus clustering)[b] | Support for prediction |
|---|---|---|---|---|---|---|
| MS0684[c] | 6q25.2-q.25.3del (3.77 Mb) | N/A | N/A | Pathogenic | 0.99 (CSS1) | N/A |
| MS0685[c] | 6q25.2-q26del (6.7 Mb) | N/A | N/A | Pathogenic | 0.98 (CSS3-NCBRS) | N/A |
| MS0686[c] | 6q24.3-q25.3del (10.3 Mb) | N/A | N/A | Pathogenic | 0.96 (CSS3-NCBRS) | N/A |
| MS0687[c] | 6q25.2-q27del (13.81 Mb) | N/A | N/A | Pathogenic | 0.95 (CSS3-NCBRS) | N/A |
| MS0672[d] | *ARID1B* (c.5833T>C, p.Cys1945Arg) | Deleterious | 0 | VUS | 0.88 (CSS1) | h, i |
| MS1220[d] | *ARID1A* (c.6507_6509dup, p.Leu2171dup) | N/A | 0 | VUS | 0.94 (CSS1) | h |
| MS1227[d] | *SMARCA2* (c.2647C>G, p.Pro883Ala) | Deleterious | 0 | VUS[g] | 0.83 (CSS3-NCBRS) | h, i, j |
| MS1185[e] | *SMARCA2* (c.2261G>C, p.Gly754Ala) | Deleterious | 0 | Likely pathogenic | 0.71 (CSS3-NCBRS) | h, i |
| MS1161[e] | *SMARCA2* (c.1697C>T, p.Ala566Val) | Conflicting | 0 | Likely pathogenic | <0.01 | k |
| MS1188[f] | *ARID1B* (c.6634C>T, p.Arg2212Cys) | Deleterious | 0.0001 | VUS | 0.04 | l, m |
| MS1189[f] | *ARID1B* (c.725G>C, p.Gly242Ala) | Conflicting | 0 | VUS | <0.01 | l |
| MS1190[f] | *ARID1B* (c.1183G>A, p.Ala395Thr) | Benign | 0 | VUS | <0.01 | l |
| MS1191[f] | *ARID1B* (c.1318T>C, p.Tyr440His) | Conflicting | 0.0004 | VUS | <0.01 | l, m |
| MS1192[f] | *ARID1B* (c.2414A>G, p.Tyr805Cys) | Conflicting | 0.001 | VUS | <0.01 | l, m |
| MS1193[f] | *ARID1B* (c.3583C>T, p.Pro1195Ser) | Conflicting | 0.0004 | VUS | <0.01 | l, m |
| MS1196[f] | *ARID1B* (c.504C>A, p.Gly168=) | N/A | 0.001 | VUS | 0.01 | l, m, n |
| MS1197[f] | *ARID1B* (c.5179G>A, p.Asp1727Asn) | Conflicting | 0 | VUS | <0.01 | l |
| MS1198[f] | *ARID1B* (c.5510A>G, p.Lys1837Arg) | Conflicting | 0 | VUS | <0.01 | l |
| MS1199[f] | *ARID1B* (c.5586A>G, p.Lys1862=) | N/A | 0.0002 | VUS | <0.01 | l, m, n |
| MS1200[f] | *ARID1B* (c.6032A>T, p.Glu2011Val) | Deleterious | 0 | VUS | <0.01 | l |
| MS1214[f] | *ARID1A* (c.1175C>A, p.Pro392His) | Deleterious | 0.0002 | VUS | 0.06 | l, m |
| MS1187[f] | *ARID1A* (c.5270C>T, p.Ala1757Val) | Benign | 8.68E−05 | VUS | 0.17 | l, m |

N/A: not applicable

[a]In silico assessment for the suspected variant was performed using three tools, SIFT, PolyPhen, and MutationTaster. A benign or deleterious decision was assigned only if all three tools were in agreement with regards to the variant. As seen, most of the predictions for the tested variants are conflicting and provide little evidence for the functional consequence of the sequence change

[b]Consensus clustering was only performed if a patient was scored high for CSS/NCBRS

[c]The first four samples were known cases of the Chr6q25del syndrome, a microdeletion involving the *ARID1B* gene

[d]These subjects were clinically suspected to be a case of CSS (MS0672 and MS1220) or NCBRS (MS1227) but with variants of unknown significance (VUS) at the time of assessment

[e]The initial clinical evaluations of subjects MS1185 and MS1161 was not available despite their reported variants being classified as likely pathogenic according to the ACMG guidelines. The classifications were made according to the existence of one strong (de novo inheritance), one moderate (absent in population databases), and one supportive (missense variant in a gene with low rates of benign missense variants—*SMARCA2*) criterion. The clinical assessment was reviewed after the DNA methylation analysis classified MS1185, but not MS1161, as a potential case of CSS/NCBRS

[f]The remaining samples were assessed due to a VUS report in a BAFopathy-related gene, and the CSS/NCBRS was not the primary clinical assumption. The objective was to screen among carriers of VUS in a CSS/NCBRS-related gene

[g]ACMG classification for this variant was later changed to likely pathogenic following the confirmation of a de novo inheritance

[h]Clinical assessments are in compliance with the suspected syndrome

[i]The suspected variant has a de novo inheritance

[j]Alternative amino acid changes are known in NCBRS at the same codon

[k]Clinical assessments are not in compliance with the suspected syndrome

[l]Non-truncating variants in *ARID1A* and *ARID1B* to cause CSS are rare

[m]Variant is present in general population at minor allele frequency above zero

[n]Variant does not induce protein change (synonymous)

performed using Illumina normalization method with background correction using the minfi package[40]. Probes with detection *p* value > 0.01, those located on chromosomes X and Y, those known to contain single-nucleotide polymorphisms at the CpG interrogation or single-nucleotide extension, and probes known to cross-react with chromosomal locations other than their target regions were removed. Arrays with more than 5% failed probe rate were excluded from the analysis. All of the samples were examined for genome-wide methylation density, and those deviating from a bimodal distribution were excluded. Factor analysis using a principal component analysis was performed to rule out batch effect or unexplained variations. Outliers were identified in this analysis and removed. No batch effect was observed by factor analysis. All of the patients and controls clustered together and did not separate based on the batch variable.

**Selection of controls for methylation profiling**. For every iteration aiming at the identification of the methylation profiles or feature selection, a random sample of controls was selected separately from our reference cohort to be compared with the case groups. Matching was done by age and array type using the MatchIt package for every group. The sample size of the controls was increased until both the matching quality and sample size were at their optimum and consistent across all analyses. This led to the determination of a control sample size six times larger than the case groups in every comparison. Increasing the sample size beyond this value impaired the matching quality. Given the CSS samples were assayed using two different Infinium arrays (450k and EPIC), we matched the controls to the cases according to the array type to ensure that our findings are not confounded by technical variation. The analyses only considered probes shared by both arrays to maintain consistency in computational workflow and biological interpretability of the findings.

**Identification of the CSS/NCBRS-related methylation profiles**. The analysis was performed using a modification of our previously published protocol[13–18,41]. The methylation level for each probe was measured as a beta value, calculated from the ratio of the methylated signals vs. the total sum of unmethylated and methylated signals, ranging between 0 (no methylation) and 1 (full methylation). This value was used for biological interpretation and visualization. For statistical analysis, wherever a normal distribution was required (linear regression modeling),

beta values were logit transformed to M-values using the following equation: log2 (beta/(1−beta)). A linear regression modeling using the limma package[42] was used to identify the differentially methylated probes. The analysis was adjusted for sex and blood cell type compositions, estimated using the minfi package according to the algorithm developed by Houseman et al.[43]. The estimated blood cell proportions were added to the model matrix of the linear models as confounding variables. The generated *p* values were moderated using the eBayes function in the limma package and were corrected for multiple testing using the Benjamini and Hochberg method. Probes with a corrected *p* value < 0.05 and a methylation difference greater than 10% were considered significant. The effect size cut-off of 10% was chosen to avoid reporting of probes with low effect size, those influenced by technical or random variations, and those not biologically interpretable, as conducted in our previous studies[13–18,41].

**Clustering analyses**. The identified probes were examined using an unsupervised hierarchical clustering analysis to ensure their ability in separating the patients from controls and to examine the similarity between various subtypes of CSS/NCBRS. Hierarchical clustering was performed using Ward's method on Euclidean distance by the gplots package. Consensus clustering was performed using ConcensusClusterPlus[44] to determine subgroups within CSS/NCBRS profile. This algorithm identifies clusters by calculating the consensus across multiple runs of a clustering algorithm and measuring the stability of the clusters. We performed 1000 clustering analyses using the *k*-means algorithm on Spearman's distance following 80% subsampling of both the samples and probes for specified cluster counts (*k*) between 2 and 10. The cluster count generating consensus cluster value >0.9 for all of the identified clusters and with the smallest PAC measure[19] was regarded the optimal *k*.

**Identification of the differentially methylated regions**. To identify genomic regions harboring methylation changes (DMRs), the DMRcate algorithm[20] was used. Using the kernel smoothing estimation of the association signals calculated for every CpG probe across the genome against a null distribution, generated using the limma regression modeling, DMRcate identified regions with a minimum of three probes no more than 1 kb apart and an average regional methylation difference >10%. We selected regions with a Stouffer transformed FDR values of the

DMR probes <0.05. The analysis was performed on the same sets of cases and controls used for methylation profiling.

**Gene-set enrichment analyses**. Gene-set enrichment analysis was performed using the missMethyl package to identify the GO terms overrepresented in the genes harboring differentially methylated probes[45], taking into account the number of CpG sites per gene. The entire CpG sites tested in the analysis were included as the background for the enrichment analysis. The enriched GO terms with an FDR < 0.05 were reported. The redundant GO terms were reduced and visualized as interactive networks using the REViGO tool[46].

**Construction of a classification model for CSS/NCBRS**. To design a classification model for CSS/NCBRS, we selected a random 75% subset of the patients with BAFopathy combined with an age-matched cohort of controls with a sample size six times larger than the patients. Feature selection was performed in two steps. First, a multivariate limma regression modeling was conducted to prioritize the differentially methylated probes which were not confounded by batch structure or blood cell type composition (incorporated into the model matrix of the regression analysis as confounding variables). The analysis was restricted to probes showing a minimum detectable methylation range by microarray technology between the cases and controls (i.e., 5%). The CpGs were sorted according to the moderated $p$ values (ascending), and the top 1000 were retained for feature selection. In the second step, we performed a receiver's operating curve characteristics analysis for every probe and identified those with an AUC > 0.85. Finally, we measured the pairwise correlations across the probes and removed the highly correlated features with $R^2 > 0.85$. The remaining probes were used to train a SVM with radial basis function kernel using e1071 package. To determine the best hyperparameters (cost and gamma), and to measure the accuracy of the model, a 10-fold cross-validation was performed. In this process, the training set was randomly divided into 10 folds. Nine folds were used for training the model and one fold for testing. After repeating this iteration for all of the 10 folds, the mean accuracy was calculated, and the hyperparameters with the optimal performance were selected. For every sample, the model was set to generate a classification score between 0 and 1 as the probability of having a methylation profile related to CSS/NCBRS. This was performed according to the Platt's scaling method[47]. The final model was applied to the testing dataset to make sure of the success of the training. The default binary classifier's probability score of 0.5 was applied as the classification cut-off.

**Validation of the classification model**. We ensured that the model is not sensitive to the batch structure of the methylation experiment by applying it to all of the samples assayed on the same batch as the cases used for training. To confirm that the classifier is not sensitive to the blood cell type compositions, we downloaded methylation data from isolated cell populations of healthy individuals from GEO (GSE35069)[21] and supplied them to our model for prediction, and examined the degree to which the scores were varied across different blood cell types. Next, the model was applied to the testing cohort (25% subset of the patients not used for feature selection or training) to evaluate the predictive ability of the model on affected subjects. To determine the specificity of the model, we supplied three groups of healthy subjects to the model for scoring: (1) healthy subjects with a diverse ethnicity background from our internal dataset, (2) publicly available methylation data from children (GSE104812)[22], and (3) publicly available methylation data from adults (GSE67705)[23]. To understand whether this model was sensitive to other medical conditions presenting with developmental delay and intellectual disabilities, we tested a large number of subjects with a confirmed clinical and molecular diagnosis of various syndromes (Fig. 8) including patients with imprinting defect disorders (Angelman syndrome ($n = 14$), Prader–Willi syndrome ($n = 7$), Silver–Russell syndrome ($n = 64$), and Beckwith–Wiedemann syndrome ($n = 9$)), diseases of the epigenomic machinery (Sotos syndrome ($n = 9$), Floating–Harbor syndrome ($n = 18$), Claes–Jensen syndrome ($n = 10$), Genitopatellar syndrome ($n = 3$), Coffin–Lowry syndrome ($n = 10$), ATRX syndrome ($n = 19$), autosomal dominant cerebellar ataxia with deafness and narcolepsy ($n = 5$), Kabuki syndrome ($n = 24$), CHARGE syndrome ($n = 39$), Rett syndrome ($n = 16$)), RASopathies (Noonan syndrome ($n = 69$), Cardiofaciocutaneous syndrome ($n = 15$), LEOPARD syndrome ($n = 2$)), Fragile X syndrome ($n = 50$), trisomy 21 ($n = 7$), and autism spectrum disorders ($n = 141$).

**Screening of undiagnosed and uncertain cases**. The finally confirmed model was used to score subjects suspected of having a CSS/NCBRS-related phenotype either with no sequence information available or with variants of unknown significance, as well as those with an uncertain clinical diagnosis. In addition, we used the model to screen a large group of individuals with various forms of DD/ID but no established diagnosis in a search for potential cases of CSS/NCBRS. The subjects who were predicted as CSS/NCBRS were evaluated based on both the clinical and molecular information. We performed in silico analysis to provide support for the predictions using SIFT, PolyPhen, and MutationTaster[48–50].

**Exome sequencing and copy number variant detection**. Subjects with no mutation report, predicted herein as a case of CSS/NCBRS, were sequenced using a custom designed next-generation sequencing panel covering coding sequences of ~4600 genes developed using SeqCap EZ MedExome probes. These genes are identified to be medically relevant by the consortium of the Emory Genetics Lab, Harvard Laboratory of Molecular Medicine, and Children's Hospital of Philadelphia. Among the CSS/NCBRS-related genes, six are covered by this panel: *ARID1A*, *ARID1B*, *ARID2*, *SMARCB1*, *SMARCA4*, and *SMARCA2*. To rule out that the patients do not have other conditions explaining their clinical presentations, we did not limit the analysis to these six genes and evaluated the entire set of the 4600 genes for a potentially causative variant. Following mapping the raw sequence data by Burrows-Wheeler aligner, VarDict was used for calling single nucleotide variants and small to moderate size indels[51]. Detection of large deletions, duplications, and insertions were performed using Pindel[52]. In addition, we investigated the copy number changes in the size of an exon and above using the ExomeDepth algorithm[53], which is shown to have an outstanding performance in the detection of rare CNVs involved in Mendelian disorders[54].

**Ethics statement**. The study protocol has been approved by the Western University Research Ethics Boards (REB ID 106302), and the Hamilton Integrated Research Ethics Board (REB ID 13-653-T). All of the participants provided informed consent prior to sample collection. All of the samples and records were de-identified before any experimental or analytical procedures. The research was conducted in accordance with all relevant ethical regulations.

## Data availability

DNA methylation microarray data from patients with CSS and NCBRS can be obtained from Gene Expression Omnibus (GEO) with accession number GSE116992 . The exome sequencing data analyzed during the current study are not publicly available as part of the conditions of the research ethical approval of the study, but are available from the corresponding author on reasonable request.

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

## Acknowledgements

We thank the staff, molecular geneticists, and clinical geneticists at London Health Science Centre, Greenwood Genetic Center, and Care4Rare Canada Consortium for identification, evaluations, and diagnosis of the patients with neurodevelopmental conditions presented in this study. We also thank the families of the patients for providing consent and information about these patients. E.A.-E. was supported by the Children's Health Research Institute Epigenetics Trainee Award, funded by the Children's Health Foundation, London, Ontario, Canada. Dedicated to the memory of Ethan Francis Schwartz, 1996–1998.

## Author contributions

E.A.-E. performed DNA methylation microarray experiment as well as statistics, machine learning and bioinformatics analysis, and wrote the paper and created images and tables. E.G.B., R.L.H., L.C.S., D.A.C., R.C., S.C.S.N., S.W.C., and C.E.S. assisted with patient recruitment, data collection, genetic variant classifications, and manuscript writing. P.M. C., C.P., V.M.S., L.B., M.A.T., D.J.C., A.M.I., S.M.W., W.S.M., A.Y.S., D.E.B., R.C.H., and K.M.B. performed phenotypic evaluations of the patients, made the clinical diagnoses, and participated in manuscript writing. G.P. assisted with generation of part of the methylation data, and manuscript writing. P.J.A., D.I.R., H.L., and C.E.S. assisted with data interpretations and manuscript writing. B.S., as the principal investigator, supervised and oversaw all aspects of this study including patient recruitment, experimental design, data analysis, and manuscript generation.

## Additional information

**Competing interests:** The authors declare no competing interests.

