## [Peer Review File · Nature Communications]

Reviewer #2 (Remarks to the Author):

This is an interesting work that rests on the hypothesis that patients with constitutional genetic alterations of the BAF complex will have impaired DNA methylation of a unique sort, detectable in the profile of DNA methylation as assessed in blood leukocytes.

The analysis initially seeks to define loci with a greater than 10% difference in methylation (I assume this is the beta value) after adjustment for blood type composition. ? There is no rationale given for the choice of 10% and one wonders (especially after examining the Table S1) if this was done post-hoc. There is no indication of how the adjustment for cell type was done; this should be explicitly included and referenced. It is unclear how the methylation data were normalized and how the authors controlled for batch effect (other than using the PCA). Was the data normalized?

Figure one is interesting and provides compelling data suggesting that the BAFopathies segregate differently in clustering. The inclusion of subjects with other "DD/ID" is apparently intended to control for "batch" effect; this is unnecessary if the data is properly normalized and given that we do not have any information on the nature of the genotype or phenotype of these additional patients, they are very poor controls and should be removed.

It would be interesting to compare the probes that were chosen for the three different conditions and this appears to be the intent of Tables S1-3; however, there appears to be no Suppl Table 2 or 3 and so this is not possible. In any case, the authors should comment on potential overlap of these lists of probes. I cannot locate Table S4, 5, 6, 7 or 8. This makes review of this manuscript difficult.

The remainder of the paper makes a reasonable case that this methylation profile has some clinical utility. This is possibly true in problematic cases where sequencing has been ambiguous or its interpretation is difficult.

In trying to understand the nature of the altered methylation profile, it would be extremely interesting to see if the same profile exists in other tissues/cells in these patients. Is it possible for the investigators to examine buccal cells in any of these patients? If the DNA methylation profile is similar it would provide compelling data addressing the hypothesis of this work.

The Figures do not have labels to indicate which Figure is which; it is fine to separate the legend but the labels on each would make it easier to figure out which is which.

Reviewer #3 (Remarks to the Author):

Authors deal with epi-signature analysis using peripheral blood DNA methylation in individuals with two BAFopathies, Coffin-Siris syndrome (CSS) and Nicholaides-Baraitser syndrome (NCBRS). They showed characteristics of epi-signature in CSS subtypes and NCBRS. Interestingly some CSS subtypes and NCBRS are more similar than that within CSS subtypes. It is also interesting that 6q 25 deletion involving ARID1B exhibits a similar CSS/NCBRS epi-signature. And the possibility was presented of clinical application using their epi-signature analysis to validate VUSs in unsolved CSS and DD/ID patients. Overall their trial looks promising. However, this reviewer feels that several points should be addressed.

1. Numbers of patients with CSS subtypes and NCBRS are significantly biased. CSS1 (ARID1B mutations): 16, CSS3 (SMARCB1 mutation): 5, CSS4 (SMARCA4 mutation): 2, NCBRS (SMARCA2 mutation): 7. Authors chose three probe sets of 146, 135 and 356 CpG sites within a minimum 10% methylation difference and a multiple-testing corrected p -value < 0.05 , but they could not find any probes for CSS4 due to the insufficient number. This reviewer thinks huge biased numbers of CSS subgroups and NCBRS might lead to significantly biased results. For example, if selected probes in CSS1 are more reliable than those of the other two groups, the cluster of CSS1 cases become solid, and CSS3/NCBRS cases are regarded as non-CSS1 cases. It seems insufficient for this reviewer to be convinced for the strong similarity between CSS3 and NCBRS. Therefore perhaps the same analysis using 5 each in CSS1, CSS3, NCBRS (of course, more is better) is recommended (at least tried).

Or even 5 in CSS4 are encouraged, though this reviewer thinks it is not easy to collect such patients.

2. In figure 2, CSS2 is seen, but this reviewer thinks CSS3. And coloring of each category is not easily differentiated. Different colors or different ways of displaying should be considered.

3. In the legend of Fig.1B, the description 146 probes (position 725 in page 27) should be 135?

4. c.1697C>T in SMARCA2 is described as de novo (page 8). Based on this epi-signature, this variant seems to be likely benign. This reviewer thinks this epi-signature system is better than the current variant classification system?

5. The 2-month-old male with score 0.86 seems to lack causative variants. Was copy-number variation completely excluded? It is well known that CMA can not detect small CNV like single exon deletion.

6. It would be kind to describe the cost aspect of epi-signature analysis if authors consider its clinical application.

7. It may be interesting to add patients with SMARCA2 duplication.

Reviewer #4 (Remarks to the Author):

Aref-Eshghi et al

The manuscript by Aref-Eshghi et al demonstrates that DNA methylation in lymphocytes is effected by mutations in subunits of the BAF complex that produce neurologic diseases including CSS and NBS in the patient. The predictive signature that they describe is probably the result of the fact that the subunits that when mutated produce neurologic disease also function in peripheral lymphocytes. Hence, one might expect such a signature, although it is surprising how robust it is.

The studies are clearly described and methods and technical approaches are pretty standard. The machine learning alorythm is interesting and broadly useful.

There is one serious mistake in the manuscript that must be removed or corrected. The authors have examined the sites of DNA methylation lymphocytes for cis associations with BAF peaks over the genome of Hela cells. This is completely wrong. They must use CHIP-seq data bases from the same lymphocyte populations that they use for the methylation analysis. BAF peaks are highly cell type specific and one would not expect that BAF peaks in Hela cells to predict function in lymphocytes. In addition, most Hela cells do not have intact BAF complexes and removing Brg in Hela cells has had no effect in the hands of many workers, although this has not been published. The authors must remove this data or say clearly that there is no reason to conclude that BAF does not work in cis.

Reviewer #2 (Remarks to the Author):

This is an interesting work that rests on the hypothesis that patients with constitutional genetic alterations of the BAF complex will have impaired DNA methylation of a unique sort, detectable in the profile of DNA methylation as assessed in blood leukocytes.

The analysis initially seeks to define loci with a greater than 10% difference in methylation (I assume this is the beta value) after adjustment for blood type composition. ? There is no rationale given for the choice of 10% and one wonders (especially after examining the Table S1) if this was done post-hoc.

We have described the rationale for choosing 10% cut-off in Methods, under the section “Identification of the CSS/NCBRS-related methylation profiles”:

“The effect size cutoff of 10% was chosen to avoid reporting of probes with low effect size, those influenced by technical or random variations, and those not biologically interpretable, as conducted in our previous studies (refs 13-18).”

In practice, microarray technology is not sensitive to detect small variations in methylation levels <5-10% and any such observed changes are susceptible to technical variations. This cut-off of 10% is a common standard for these studies and is recommended by other software packages designed for methylation analysis including “metilene” and “BSmooth”:

Jühling F, Kretzmer H, Bernhart SH, Otto C, Stadler PF, Hoffmann S. metilene: fast and sensitive calling of differentially methylated regions from bisulfite sequencing data. *Genome research*. 2016;26(2):256-62.

Hansen KD, Langmead B, Irizarry RA. BSmooth: from whole genome bisulfite sequencing reads to differentially methylated regions. *Genome biology*. 2012;13(10):R83.

There is no indication of how the adjustment for cell type was done; this should be explicitly included and referenced.

We performed an adjustment for blood cell type composition using limma multivariate regression modeling. Cell type composition was estimated using minfi package. We have referenced minfi package in the manuscript, which uses the algorithm by Houseman et al. 2012 for blood cell type estimation. We have now added the reference of this algorithm to the manuscript. The estimated cell proportions were incorporated into the model matrix of the multivariable regression model. Therefore the beta values that are presented here are not corrected. Instead, the p-value is adjusted for variations in the blood subtypes. Using this method we ensured that none of the identified changes are caused by variations in blood cell types. As suggested by the reviewer we have clarified this point in Methods, under the section “Identification of the CSS/NCBRS-related methylation profiles”:

“The analysis was adjusted for sex and blood cell type compositions, estimated using the minfi package according to the algorithm developed by Houseman et al. The estimated blood cell proportions were added to the model matrix of the linear models as confounding variables.”

In addition to the above described method, we evaluated the performance of our machine learning model on different cell populations to ensure any variation by blood composition does not influence the classification outcomes. This is presented in the results under the section “Construction and validation of a classification model for CSS/NCBRS”:

“we evaluated the extent to which this model is sensitive to variations in blood cell type compositions by applying it to a total of 60 methylation array data files from six healthy individuals, downloaded from gene expression omnibus (GEO, GSE35069) , each being assayed for whole blood, peripheral blood mononuclear cells (PBMC), and granulocytes, as well as for seven isolated cell populations (CD4+ T, CD8+ T, CD56+ NK, CD19+ B, CD14+ monocytes, neutrophils, and eosinophils), separately. All of these samples were classified as non-CSS/NCBRS with scores ranging 0.01-0.13 (Table S8), consistent with those generated for the whole blood samples. The average inter-cell-type variability in the scores was determined to be not more than 10% (Table S8).

It is unclear how the methylation data were normalized. how the authors controlled for batch effect (other than using the PCA). Was the data normalized?

The data was normalized as indicated in the Methods, “Methylation assay and quality control”:

“Normalization was performed using Illumina normalization method with background correction using the minfi package”

We controlled for batch effect at five stages after normalization:

- 1- At the stage of sample selection, given the patients were assayed using two platforms, we matched the cases and controls by the platform variable to avoid any bias induced by the array type. This variable would be expected to have most potential for batch effects, which we addressed by using a matching approach. Methods, Selection of controls for methylation profiling:

“Given the CSS samples were assayed using two different Infinium arrays (450k and EPIC), we matched the controls to the cases according to the array type to ensure that our findings are not confounded by technical variation. The analyses only considered probes shared by both arrays, to maintain consistency in computational workflow and biological interpretability of the findings.”

- 2- We performed a factor analysis using PCA to ensure that the samples do not segregate based on batch variable. We did not observe a separation based on batch. All of the cases and controls clustered together. Methods, Methylation assay and quality control:

“Factor analysis using a principal component analysis was performed to rule out batch effect or unexplained variations. Outliers were identified in this analysis and removed. No batch effect was observed by factor analysis. All of the patients and controls clustered together and did not separate based on the batch variable.”

- 3- At the analysis stage, the identified probes were used to see whether they can differentiate the samples on the same batch as the cases from the controls using a hierarchical clustering, which is presented in figure 1:

“To assure that these observations are not the result of a general DD/ID methylation pattern or batch effect, a random mix of non-CSS/NCBRS subjects having other forms of DD/ID, assayed on the same batch as the patients, were included in each clustering—none grouped with the CSS/NCBRS samples (Figure 1).”

- 4- At the stage of feature selection for training a classification model for CSS/NCBRS, since we had to combine multiple samples, which required the use of data from several batches, we added the batch variable as a confounding factor to the regression analysis to prioritize probes that are not influenced by batch effect. Methods, Construction of a classification model for CSS/NCBRS:

“multivariate limma regression modeling was conducted to prioritize the differentially methylated probes which were not confounded by batch structure or blood cell type composition (incorporated into the model matrix of the regression analysis as confounding variables).”

- 5- Finally, after the model was developed we tested if it can differentiate between the cases and the other samples assayed on the same batch. Results, Construction and validation of a classification model for CSS/NCBRS:

“Next, we confirmed that the model could completely differentiate the non-CSS/NCBRS samples assayed on the same technical batch of the cases (n=244) from the true CSS/NCBRS samples by supplying their methylation values into the model for prediction. None of these samples were scored high for CSS/NCBRS, indicating the trained model is not sensitive to the batch structure of the data.”

Figure one is interesting and provides compelling data suggesting that the BAFopathies segregate differently in clustering. The inclusion of subjects with other “DD/ID” is apparently intended to control for “batch” effect; this is unnecessary if the data is properly normalized and given that we do not have any information on the nature of the genotype or phenotype of these additional patients, they are very poor controls and should be removed.

This analysis was done to control that our observation is not due to batch effect or a pattern shared by all DD/IDs. We agree that this might be confusing. Therefore we have transferred that figure to the supplementary material, and generated new figures as recommended by the reviewer excluding the DD/ID cohort. Also figure 1C is now presented as figure 2 to improve the illustration quality.

It would interesting to compare the probes that were chosen for the three different conditions and this appears to be the intent of Tables S1-3; however, there appears to be no Suppl Table 2 or 3 and so this is not possible. In any case, the authors should comment on potential overlap of these lists of probes. I cannot locate Table S4, 5, 6, 7 or 8. This makes review of this manuscript difficult.

Tables S2-8 are located on sheets (tabs) 2-8 of the same excel file as the one containing table S1 (to reduce the number of separate supplemental files). The nominal count of probes shared (overlapped) across the three lists is negligible, ranging 10-20%. We have now added the information regarding the nominal overlap into the Results, CSS/NCBRS exhibit a DNA methylation epi-signature spectrum:

“A nominal overlap of 10-20% was observed across the three sets”.

“These observations indicated that, the identified CpG probes are not entirely specific to every BAF subunit and there exists a spectrum of methylation change shared across different subtypes.”

The remainder of the paper makes a reasonable case that this methylation profile has some clinical utility. This is possibly true in problematic cases where sequencing has been ambiguous or its interpretation is difficult. In trying to understand the nature of the altered methylation profile, it would

be extremely interesting to see if the same profile exists in other tissues/cells in these patients. Is it possible for the investigators to examine buccal cells in any of these patients? If the DNA methylation profile is similar it would provide compelling data addressing the hypothesis of this work.

We agree with the reviewer that this question would have been of interest to investigate. However, we currently have no data from other tissue sources. Previous studies on Sotos syndrome has shown that a methylation profiles can be present in other tissues including fibroblast. However, this might not necessarily be applicable to BAFopathies. We have now added a new paragraph to the end of discussion to discuss future directions of this study and other points raised by reviewers:

“Alongside unraveling the functional continuum and diagnostic utility of an epigenetic signature in BAFopathies our findings generate new questions to be explored by further research. We have not investigated whether the observed changes are present in other organs, particularly in the neurological tissues which contribute to majority of clinical presentation in these conditions. Our previous study has shown that a hypomethylation profile detected in the blood of the Sotos syndrome can also be found in the fibroblasts of the patients. However, we acknowledge that this might not be the case for BAFopathies and warrants further investigation. While providing foundation for hypothesis generation of functional consequences of these epigenetic changes, the observed methylation signatures and their impact on the pathology of the disease remain to be investigated. These may include integrative investigation of gene expression changes and histone modification patterns among the affected individuals. Animal models to study development can reveal the origin of the identified changes and potentially shed light on the pathways involved in pathogenesis. Other less common and yet to be identified gene mutations associated with BAFopathy syndromes will need to be investigated. With increasing methylation databases it may be possible to examine differences in methylation patterns resulting from different mutation types, including loss and gain of function mutations and deletion/duplications in genes encoding different proteins of the BAF complex.”

The Figures do not have labels to indicate which Figure is which; it is fine to separate the legend but the labels on each would make it easier to figure out which is which.

We apologize for the inconvenience. We followed the Journal’s regulation when submitting the files which were compiled/assembled by the submission portal.

Reviewer #3 (Remarks to the Author):

Authors deal with epi-signature analysis using peripheral blood DNA methylation in individuals with two BAFopathies, Coffin-Siris syndrome (CSS) and Nicholaidis-Baraitser syndrome (NCBRS). They showed characteristics of epi-signature in CSS subtypes and NCBRS. Interestingly some CSS subtypes and NCBRS are more similar than that within CSS subtypes. It is also interesting that 6q 25 deletion involving ARID1B exhibits a similar CSS/NCBRS epi-signature. And the possibility was presented of clinical application using their epi-signature analysis to validate VUSs in unsolved CSS and DD/ID patients. Overall their trial looks promising. However, this reviewer feels that several points should be addressed.

1. Numbers of patients with CSS subtypes and NCBRS are significantly biased. CSS1 (ARID1B mutations): 16, CSS3 (SMARCB1 mutation): 5, CSS4 (SMARCA4 mutation): 2, NCBRS (SMARCA2 mutation): 7. Authors chose three probe sets of 146, 135 and 356 CpG sites within a minimum 10% methylation difference and a multiple-testing corrected p-value<0.05, but they could not find any probes for CSS4 due to the

insufficient number. This reviewer thinks huge biased numbers of CSS subgroups and NCBRS might lead to significantly biased results. For example, if selected probes in CSS1 are more reliable than those of the other two group, the cluster of CSS1 cases become solid, and CSS3/NCBRS cases are regarded as non-CSS1 cases. It seems insufficient for this reviewer to be convinced for the strong similarity between CSS3 and NCBRS. Therefore perhaps the same analysis using 5 each in CSS1, CSS3, NCBRS (of course, more is better) is recommended (at least tried). Or even 5 in CSS4 are encouraged, though this reviewer thinks it is not easy to collect such patients.

We thank the reviewer for pointing this important point out. As suggested, to address the question of the possible bias due to differences in sample sizes we performed the analysis for CSS1 and NCBRS by randomly selecting 5 samples from each; so that all findings are obtained from 5 cases and 30 controls for all three categories (sample size for CSS3 was already 5 in the study). As expected, reducing the sample sizes results in reduction in detected probe counts from 146 and 365 to 27 and 157 for CSS1 and NCBRS, respectively. Next, the clustering analysis is conducted. Interestingly, the 27 probes perform just as well as the 146 probes in placing all BAFopathies into one cluster. The 157 probes also generate the same exact pattern observed by 365 probes in figure 2, i.e., clustering controls and CSS1 into one cluster, CSS3 and NCBRS into the other, and splitting the CSS4 in between the two. Next, we perform a consensus clustering, same as what was conducted in the manuscript on the combination of the three probes sets for CSS1, CSS3, and NCBRS (n=15, five each). The result is exactly the same as the one presented in the manuscript. We also questioned whether this observation could be due to differences in probe counts, since the CSS3 and NCBRS have ~150 probes vs. CSS1 has only 27. To control for this, we selected the top 150 probes sorted by p-value from the comparison of 5 CSS1 cases and 30 controls, and reassessed the data. The conclusion remained the same.

This entire analysis is now presented in supplementary figures S19-21. The similarity between CSS3 and NCBRS is also supported from the analysis of differentially methylated regions in BAFopathies (Figures 4-6).

Figure S19- To address the question of whether the relatively large CSS1 sample size has resulted in the generation of a more reliable probe list, and thus, lack of distinction across different CSS/NCBRS subtypes. We randomly selected five CSS1 cases and compared them with 30 matched controls, exactly as conducted in the manuscript. The sample size of five was chosen to make the sample count equal to that in CSS3 (five cases and 30 controls). This analysis identified 27 probes which perform equally well as the initial 146 CSS1 probes in placing all BAFopathies into one cluster, indicating that our initial observation of CSS1 having a profile that is shared across all BAFopathies is not influenced by the relatively larger sample size of CSS1.

Figure S20- A similar analysis to that in Figure S19 was performed for NCBRS cases. As expected, the identified probe count was reduced to 157 from the initial 365. However, this did not change their performance in the clustering analysis, and similar to the initially identified probes, this experiment clustered controls and CSS1 into one cluster, CSS3 and NCBRS into the other, and split the two CSS4 subjects between the two.

Figure S21- Using the combination of the identified probes in the analysis of Figures S19 and S20 as well as the 135 probes found in CSS3, a consensus clustering was performed, as explained previously in Figure 3. This was only conducted for CSS1, CSS3, and NCBRS ($n=15$, five each) to ensure equal sample sizes in each three categories. The same result as the one in Figure 3 was observed, i.e., CSS3 and NCBRS cases always cluster together, whereas CSS1 generates a completely separate cluster of the two (section A). We next questioned whether this observation could be due to differences in probe counts, since the CSS3 and NCBRS have ~ 150 probes, whereas CSS1 has only 27. To control for this, we selected the top 150 probes sorted by p-value from the comparison of 5 CSS1 cases and 30 controls, and re-performed the analysis (section B). The same observation was noted, indicating that the similarity observed between CSS3 and NCBRS is not related to probe count or the sample size of the three subtypes.

2. In figure 2, CSS2 is seen, but this reviewer thinks CSS3. And coloring of each category is not easily differentiated. Different colors or different ways of displaying should be considered.

Upon careful review, we believe the labeling error that reviewer correctly points out is in figure 1. We have now fixed this error. For improving the quality, figure 1c is now presented as a separate figure (figure 2). As well, the samples for batch control are removed as per request by the other reviewer (transferred to supplementary data), which has improved the visualization. Also, we enhanced the colors to improve visual distinctions. The font and quality of all figures can be further modified in consultation with the graphists of Nature Communications.

3. In the legend of Fig.1B, the description 146 probes (position 725 in page 27) should be 135?

This is a typo. We thank the reviewer for detecting this error. We have corrected that number to 135.

4. c.1697C>T in SMARCA2 is described as de novo (page 8). Based on this epi-signature, this variant seems to be likely benign. This reviewer thinks this epi-signature system is better than the current variant classification system?

We believe our method should be used in conjunction with the variant classification system. They perform different assessments which may complement each other in some cases as well as provide differentiating information in others. Classification system evaluates the variant, whereas our method provides an assessment of the functional disruption of the BAF complex. We have emphasized this point and added following to the discussion:

“An example was the subject with a de novo missense variant in SMARCA2 (Table 2), where strict adherence to the variant classification system had resulted in a likely pathogenic classification. This patient was found later to not show clinical features consistent with NCBRS and was scored 0.01 by our DNA methylation model. This indicates that the reported variant is benign and sequence variant assessment alone without consideration of clinical and functional data can sometimes be misleading.”

5. The 2-month-old male with score 0.86 seems to lack causative variants. Was copy-number variation completely excluded? It is well known that CMA can not detect small CNV like single exon deletion.

We have also examined the CNVs status of this subject using the exome CNV assessment tool and did not identify any changes. In up to 40% of patients matching Coffin-Siris clinical criteria a causative variant is not found, which may reflect yet undiscovered genetic etiology or possibility of causative variants in un-assessed non-coding gene regions. This is now added to the manuscript.

Results, last paragraph: “Analysis of copy number variations obtained from the exome data neither found a pathogenic event in a gene known to cause a DD/ID condition.”

Methods, last paragraph: “In addition, we investigated the copy number changes in the same subjects using the ExomeDepth algorithm, which is shown to have outstanding performance in the detection of rare CNVs involved in Mendelian disorders.”

6. It would be kind to describe the cost aspect of epi-signature analysis if authors consider its clinical application.

This is now discussed in the discussion:

“We propose that genomic DNA methylation assessment has a potential to become part of the clinical screening of patients with a broad range of developmental disorders, taking into consideration that, in addition to BAFopathies, it allows for concurrent assessment of multiple syndromes resulting from DNA methylation defects, including imprinting conditions, Fragile X syndrome, and number of other previously described conditions with epi-signatures. Given the cost of DNA methylation microarray testing is similar to the cost of an average single-gene molecular test, it has a potential to be adopted in molecular diagnosis together with current genomic screening tests including CNV microarray testing and exome sequencing.”

7. It may be interesting to add patients with SMARCA2 duplication.

Unfortunately, we don't have any patient with this form of mutation. Only two subjects with SMARCA2 duplications are reported in the literature (PMID: 27264538). We also think that this is worthwhile to study alongside other BAFopathies. We have discussed this along with other issues that warrant further investigations in a new paragraph at the end of the discussion as questions that remain to be investigated in the future that our work cannot answer:

“Alongside unraveling the functional continuum and diagnostic utility of an epigenetic signature in BAFopathies our findings generate new questions to be explored by further research. We have not investigated whether the observed changes are present in other organs, particularly in the neurological tissues which contribute to majority of clinical presentation in these conditions. Our previous study has shown that a hypomethylation profile detected in the blood of the Sotos syndrome can also be found in the fibroblasts of the patients. However, we acknowledge that this might not be the case for BAFopathies and warrants further investigation. While providing foundation for hypothesis generation of functional consequences of these epigenetic changes, the observed methylation signatures and their impact on the pathology of the disease remain to be investigated. These may include integrative investigation of gene expression changes and histone modification patterns among the affected individuals. Animal models to study development can reveal the origin of the identified changes and potentially shed light on the pathways involved in pathogenesis. Other less common and yet to be identified gene mutations associated with BAFopathy syndromes will need to be investigated. With increasing methylation databases it may be possible to examine differences in methylation patterns resulting from different mutation types, including loss and gain of function mutations and deletion/duplications in genes encoding different proteins of the BAF complex.”

Reviewer #4 (Remarks to the Author):

The manuscript by Aref-Eshghi et al demonstrates that DNA methylation in lymphocytes is effected by mutations in subunits of the BAF complex that produce neurologic diseases including CSS and NBS in the patient. The predictive signature that they describe is probably the result of the fact that the subunits that when mutated produce neurologic disease also function in peripheral lymphocytes. Hence, one might expect such a signature, although it is surprising how robust it is.

The studies are clearly described and methods and technical approaches are pretty standard. The machine learning algorithm is interesting and broadly useful.

There is one serious mistake in the manuscript that must be removed or corrected. The authors have examined the sites of DNA methylation lymphocytes for cis associations with BAF peaks over the genome of HeLa cells. This is completely wrong. They must use CHIP-seq data bases from the same lymphocyte populations that they use for the methylation analysis. BAF peaks are highly cell type specific and one would not expect that BAF peaks in HeLa cells to predict function in lymphocytes. In addition, most HeLa cells do not have intact BAF complexes and removing Brg in HeLa cells has had no effect in the hands of many workers, although this has not been published. The authors must remove this data or say clearly that there is no reason to conclude that BAF does not work in cis.

We absolutely agree with the reviewer about the tissue specificity of both epigenetic signatures protein bounding patterns. Therefore the comparison of DNA methylation from peripheral blood and BAF-DNA binding patterns from HeLa cells is at best speculative and does not add anything constructive to the

manuscript. Since currently, no such data set is available from blood cells, as suggested by the reviewer, we removed this analysis and discussion from the manuscript.

Reviewer #2 (Remarks to the Author):

The revision has addressed my concerns.

Reviewer #3 (Remarks to the Author):

Authors addressed almost perfectly this reviewer's concerns. Especially they reanalyzed again using similar numbers of CSS1, CSS3 and NCBRS and could obtain similar results though the size of probes was reduced as shown in figures S19-S21. Therefore this reviewer now believes their findings and thinks it important that their method may be able to complement the current technologies for the molecular diagnosis. Well done.